# Unified Diffusion VLA: Vision-Language-Action Model via Joint Discrete Denoising Diffusion Process

**Jiayi Chen**[1,*] **Wenxuan Song**[1,*,†] **Pengxiang Ding**[2,3,†] **Ziyang Zhou**[1] **Han Zhao**[2,3]
**Feilong Tang**[4] **Donglin Wang**[2] **Haoang Li** [1,‡]
[1]The Hong Kong University of Science and Technology (Guangzhou)
[2]Westlake University    [3]Zhejiang University    [4]Monash University

## Abstract

Vision-language-action (VLA) models aim to understand natural language instructions and visual observations and to execute corresponding actions as an embodied agent. Recent work integrates future images into the understanding-acting loop, yielding unified VLAs that jointly understand, generate, and act—reading text and images and producing future images and actions. However, these models either rely on external experts for modality unification or treat image generation and action prediction as separate processes, limiting the benefits of direct synergy between these tasks. Our core philosophy is to optimize generation and action jointly through a synchronous denoising process, where the iterative refinement enables actions to evolve from initialization, under constant and sufficient visual guidance. We ground this philosophy in our proposed Unified Diffusion VLA and Joint Discrete Denoising Diffusion Process (JD3P), which is a joint diffusion process that integrates multiple modalities into a single denoising trajectory to serve as the key mechanism enabling understanding, generation, and acting to be intrinsically synergistic. Our model and theory are built on a unified tokenized space of all modalities and a hybrid attention mechanism. We further propose a two-stage training pipeline and several inference-time techniques that optimize performance and efficiency. Our approach achieves state-of-the-art performance on benchmarks such as CALVIN, LIBERO, and SimplerEnv with $4\times$ faster inference than autoregressive methods, and we demonstrate its effectiveness through in-depth analysis and real-world evaluations. Our project is available at this url.

## 1 Introduction

Vision-language-action models (VLAs) (Tian et al., 2025; Kim et al., 2025; Black et al., 2024) aim to understand natural language instructions and visual observations, and to execute corresponding actions as an embodied agent. In recent years, some research (Wang et al., 2025; Lv et al., 2025) has integrated future images into the understanding-acting loop to build robust, foresight-driven policies. This paradigm confers planning capabilities to the model by predicting future images before action inference, thereby converting the abstract task of action prediction into a more tractable inverse kinematics problem. In this work, we term these models as **unified VLAs**, which understand, generate, and act across modalities—reading visual content and text and outputting visual content and actions.

To build unified VLAs, two existing popular paradigms fall short (Table 1): (1) One line of research (Wu et al., 2024; Tian et al., 2025; Zhang et al., 2025a) unifies these modalities using extrinsic experts as encoders (Dosovitskiy et al., 2020; Radford et al., 2021) and decoders. These VLAs encode images and language and output intermediate tokens, which serve as conditions for visual and action generation models (*e.g.*, diffusion models) to produce future images and actions. However, such

---

[*]Equal contribution
[†]Project lead: `songwenxuan0115@gmail.com`, `dingpx2015@gmail.com`
[‡]Corresponding author: `haoangli@hkust-gz.edu.cn`

modular separation often introduces misalignment, higher complexity, and weak coupling between visual generation and action prediction. (2) The other line (Zhao et al., 2025; Wang et al., 2025; Cen et al., 2025) unifies the input and output space through visual (Van Den Oord et al., 2017) and action tokenizers (Pertsch et al., 2025), which allows vision-language-action alignment in a token-level space and does not require extra encoders or decoders. However, in these designs, image generation and action prediction still remain separate processes, restricting the ability of the model to leverage rich future visual information for action prediction. Moreover, some of these approaches (Wang et al., 2025; Zhong et al., 2025) only predicted images during training as an auxiliary task, thereby forfeiting the value of using future images as explicit guidance at inference time. We therefore posit that *a genuine unification of understanding, generation, and acting requires these processes to be intrinsically synergistic, ensuring that actions are formulated as implicit mappings to the desired future observations.*

One potential approach is to generate image and action tokens in an autoregressive manner. However, by design, each action token only integrates contextual information through a single computation, which restricts the extent to which image generation can guide action prediction. In contrast, our core philosophy is to **optimize visual generation and action prediction jointly through a synchronous denoising process**. Here, at every denoising iteration, all action tokens causally attend to all future image tokens, and this computation is repeated multiple times across the denoising trajectory. This iterative scheme ensures that each action token is progressively refined under sufficient guidance from future visual predictions. The coarse-to-fine refinement allows actions to evolve from initialization alongside the denoising of future images, and finally converge into precise actions under a confidence-based criterion. This design effectively transforms latent visual representations into temporally structured actions.

To this end, we propose our **Unified Diffusion VLA** (UD-VLA) and **Joint Discrete Denoising Diffusion Process** (JD3P). JD3P is a joint diffusion process that integrates multiple modalities into a single synchronous denoising trajectory, which serves as the key mechanism enabling understanding, generation, and acting to mutually reinforce one another. We construct our model and theory on the foundation of two techniques. (1) We unify the multimodal space through discrete tokenization, employing a VQ-based visual tokenizer (Van Den Oord et al., 2017; Zheng et al., 2022) for images and the FAST action tokenizer (Pertsch et al., 2025) for actions. (2) We design a hybrid attention mechanism that enables rich intra-modal interactions for images and actions while enforcing causal attention across modalities, thereby preserving the nature of inverse kinematics and ensuring that action tokens are sufficiently conditioned on future visual information. For training, we conduct a two-stage pipeline to extend a pretrained VLM with image prediction capability to capture and model world dynamics in stage (i), and then apply JD3P on robot action datasets to jointly train image generation and action prediction in stage (ii). During inference, efficiency is improved through KV-cache and prefilled special tokens, candidate tokens are narrowed by remapping into a smaller range, and precision is further ensured via confidence-based decoding.

Our method achieves state-of-the-art (SOTA) performance on multiple popular benchmarks, including CALVIN (Mees et al., 2022), LIBERO (Liu et al., 2023), and SimplerEnv (Li et al., 2024b), while maintaining a $4\times$ over autoregressive methods. Furthermore, in-depth analysis verifies the effectiveness of our designs, and real-world evaluations further demonstrate practical utility. In summary, our contributions are threefold:

- We propose the unified diffusion VLA, which tightly couples understanding, generation, and acting in a mutually beneficial manner.
- We instantiate this design via discrete tokenization, hybrid attention, and the JD3P process as the central mechanism for cross-modal synergy.
- We design a two-stage training pipeline to activate the image generation capabilities and introduce several test-time techniques that ensure both high performance and efficiency.

## 2 RELATED WORKS

**Visual Prediction for Robot Manipulation.** Recent studies have incorporated visual prediction tasks to enhance robotic manipulation capabilities. One line of work aims to decompose the action manipulation task into a visual prediction task and an inverse dynamics task. Early work Wang et al.

Table 1: Component comparison across unified VLAs. WorldVLA and UniVLA model visual and action tokens in post-training, but decode only action tokens at inference (shown in gray text). "IBA" denotes bidirectional attention within the visual modality and the action modality separately, while they use causal attention across modalities. "CA-BA" denotes causal attention on visual tokens and bidirectional attention on action tokens. "CA" denotes causal attention. For the decoding process,"–" denotes separate vision and action decoding processes. "Diff" denotes a diffusion-based decoding process. "AR" denotes autoregressive decoding. See Appendix Section E for details of losses.

| Method | Visual Output | | Action Output | | Attention Mechanism | Decoding Process |
|---|---|---|---|---|---|---|
| | Loss | Expert | Loss | Expert | | |
| *Unified Modalities with Extrinsic Experts* | | | | | | |
| GR-1 (Wu et al., 2024) | $\mathcal{L}_{\text{MSE}}$ | ✓ | $\mathcal{L}_{\text{Diff-cont}}$ | ✓ | CA | Diff-Diff |
| SEER (Tian et al., 2025) | $\mathcal{L}_{\text{MSE}}$ | ✓ | $\mathcal{L}_{\text{Diff-cont}}$ | ✓ | CA | Diff-Diff |
| DreamVLA (Zhang et al., 2025b) | $\mathcal{L}_{\text{MSE}}$ | ✓ | $\mathcal{L}_{\text{Diff-cont}}$ | ✓ | CA | Diff-Diff |
| F1 (Lv et al., 2025) | $\mathcal{L}_{\text{Diff-disc}}$ | ✓ | $\mathcal{L}_{\text{Diff-cont}}$ | ✓ | IBA | Diff-Diff |
| UP-VLA (Zhang et al., 2025a) | $\mathcal{L}_{\text{Diff-disc}}$ | ✗ | $\mathcal{L}_{\text{MSE}}$ | ✓ | IBA | Diff-Diff |
| *Unified Input and Output Space in Seperate Decoding Processes* | | | | | | |
| COT-VLA (Zhao et al., 2025) | $\mathcal{L}_{\text{Diff-disc}}$ | ✗ | $\mathcal{L}_{\text{Diff-disc}}$ | ✗ | CA-BA | AR-Diff |
| WorldVLA (Cen et al., 2025) | $\mathcal{L}_{\text{NTP}}$ | ✗ | $\mathcal{L}_{\text{NTP}}$ | ✗ | CA | AR |
| UniVLA (Wang et al., 2025) | $\mathcal{L}_{\text{NTP}}$ | ✗ | $\mathcal{L}_{\text{NTP}}$ | ✗ | C | AR |
| *Unified Input, Output in A Joint Decoding Processes* | | | | | | |
| **UD-VLA (Ours)** | $\mathbf{\mathcal{L}_{\text{Diff-disc}}}$ | ✗ | $\mathbf{\mathcal{L}_{\text{Diff-disc}}}$ | ✗ | **IBA** | **Diff** |

(2019) utilizes a CIGAN Kurutach et al. (2018) to generate visual plans as a reference trajectory to track with a visual servoing controller, which is learned as an inverse dynamics model. Then, UniPi Du et al. (2023) regards the sequential decision-making problem as a text-conditioned video generation problem, which leads to a generalized, multi-task policy and naturally inherits the knowledge of video generation models. VLP Du et al. (2024) is constructed to generate long video plans, including visual and language contents, which are finally transferred into long-horizon robot actions. To address the limited generalization caused by the small scale of robot data, Luo et al. (2025) proposes Inverse Probabilistic Adaptation to adaptively integrate in-domain information into large-scale text-to-video models.

Another line of work aims to unify the image generation and action generation in a joint process. PAD Guo et al. (2024) utilizes a diffusion transformer to realize the simultaneous prediction of future images and robot actions. UVA Li et al. (2025) designs a joint sequence model to encode observations and actions into latent, and uses the latent to guide two separate denoising processes for videos and actions. While these methods unify the generation of images and actions through a continuous diffusion process on the diffusion transformer, our UD-VLA realizes the unified generation through a discrete diffusion process on the large vision-language model as a backbone, which contributes to the future research of inheriting the knowledge of the pre-trained vision-language models.

**Unified VLAs.** Recent work has increasingly adopted unified vision–language–action (VLA) architectures, as summarized in Table 1. Early methods incorporated an auxiliary objective for predicting future images, employed external encoders and decoders during both training and inference to unify image generation and action prediction. One line of work (Wu et al., 2024; Tian et al., 2025; Zhang et al., 2025b) supervises visual representation learning via reconstruction losses and learns action prediction through diffusion-based contrastive objectives under a causal attention mask in an autoregressive manner. UP-VLA (Zhang et al., 2025a) pioneers a unified VLA pre-training paradigm that jointly optimizes multimodal understanding and future visual prediction, retaining semantic and spatial detail for stronger generalization and precise control. F1 (Lv et al., 2025) integrates visual foresight into the perception–acting loop via a mixture-of-transformers with next-scale prediction, generating future visual states as planning targets and guiding actions through foresight-driven inverse dynamics. Building on advances in unified multimodal models (Wang et al., 2024; Yang et al., 2025; Deng et al., 2025), recent VLA approaches treat inputs and outputs within a shared token space, enabling native processing of all modalities and obviating external encoders or decoders.

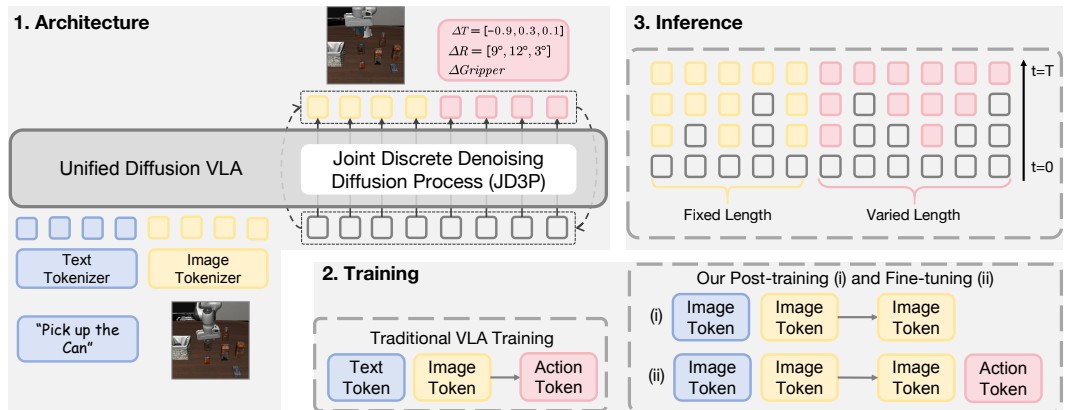

Figure 1: **Overview of our Unified Diffusion VLA.** 1. We construct our UD-VLA and formalize a Joint Discrete Denoising Diffusion Process (JD3P) to allow visual generation and action prediction to be intrinsically synergistic. 2. We design a two-stage training, including a post-training stage in a world-model manner to predict future images and a fine-tuning stage to generate both future images and actions. 3. During inference, the noising fixed-length image tokens and varied-length action tokens are denoised into clean tokens after $T$ steps in JD3P.

CoT-VLA (Zhao et al., 2025) adopts diffusion-based objectives and asymmetric attention in a unified token space, decoding vision in an autoregressive manner and actions via diffusion, thereby making the visual chain of thought explicit through subgoal images. WorldVLA (Cen et al., 2025), UniVLA (Wang et al., 2025), and FlowVLA (Zhong et al., 2025) treat both perception and control as next token under causal attention. In post-training, both visual and action tokens are modeled, while at inference only action tokens are decoded with vision serving as conditioning. However, current unified VLAs seldom unify decoding, which remains the dominant source of latency. Visual chain of thought pipelines still perform iterative image denoising and autoregressive action decoding, incurring high computational cost on both branches.

**Discrete Diffusion VLA.** PD-VLA (Song et al., 2025b), as an early model employing discrete diffusion inference, adopts a BART-style (Lewis et al., 2020) denoising strategy. In this approach, a subset of action tokens is randomly substituted with vocabulary tokens and then iteratively refined to recover the ground-truth sequence. In contrast, Discrete Diffusion VLA (Liang et al., 2025) and LLADA-VLA (Wen et al., 2025) follow the BERT-style (Devlin et al., 2019) masked prediction strategy, where selected action tokens are replaced with a special [MASK] token, and the model directly learns to predict the original tokens at these masked positions. To further improve efficiency, CEED-VLA (Song et al., 2025a) employs consistency distillation to reduce the number of iterations in the discrete diffusion process, leading to over $4\times$ speedup without compromising performance. However, these discrete diffusion VLAs focus exclusively on action prediction while largely ignoring the interplay between visual and action tokens, thus failing to fully exploit the potential benefits of cross-modal representation learning.

## 3 METHODS

We propose UD-VLA, a unified diffusion VLA that bridges vision-language understanding, future image generation, and action prediction in a single transformer. We first construct a unified multimodal space by quantizing multimodal information into discrete tokens (Section 3.1). Next, we design a hybrid attention mechanism to maximize the utilization of information from each modality. We then formalize our core theory, Joint Discrete Denoising Diffusion Process (JD3P), and reformulate the loss computation to support its training (Section 3.2). Finally, we balance performance and efficiency during inference through several key techniques (Section 3.3).

### 3.1 UNIFIED DIFFUSION VLA

**Unified Tokenization.** As illustrated in Figure 1, our method unifies language, vision, and action modalities by converting each into discrete tokens and concatenating them into a single multimodal

sequence. The language tokens follow the same design as Emu3 (Wang et al., 2024), visual observations are discretized with a VQ tokenizer (Zheng et al., 2022) into $V_v$ tokens in the vocabulary, and actions are represented using FAST (Pertsch et al., 2025) into $V_a$ tokens in the vocabulary. To explicitly structure the diffusion process for different modalities, we employ special tokens, `<BOI>` and `<EOI>` for images, and `<BOA>` and `<EOA>` for actions, to mark the beginning and ending of their respective token sequences. Then, the complete token sequences are formatted as:

$$[ \text{ text tokens } ; \text{ current image tokens } ; \text{ future image tokens } ; \text{ action tokens } ], \tag{1}$$

where text tokens and current image tokens serve as input, and future image tokens and action tokens are output.

**Hybrid Attention Mechanism.** To coordinate multimodal tokens, we introduce a hybrid attention mechanism in Figure 2. The input text and current image follow causal attention and bidirectional attention separately. We split output tokens into generation (future image) blocks and acting blocks. Within each block, bidirectional attention enables comprehensive token interactions, which further break the time-serial dependence between action tokens to avoid shortcut learning (Torne et al., 2025). Tokens across blocks are connected through causal attention: the generation block attends to the input, and the acting block attends to both, while no information flows backward. This progressive design recasts the otherwise difficult objective of end-to-end action prediction in VLMs into two coupled processes: (i) a foresight process that predicts the next visual state, and (ii) an inverse kinematics process that infers the action condi-

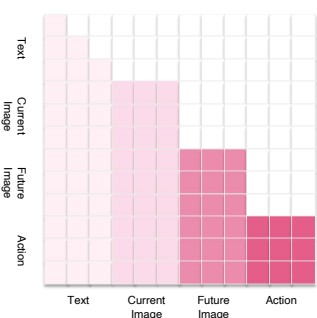

Figure 2: **Hybrid attention mechanism in UD-VLA.**

tioned on that visual prediction. This factorization enables principled, targeted training of both stages (see Section 3.2). By explicitly prohibiting action-to-vision pathways, it eliminates coarse action information leakage and the attendant error compounding, improves interpretability, and ensures that downstream control is genuinely grounded in predicted visual consequences rather than spurious correlations.

**Joint Discrete Denoising Diffusion Process (JD3P).** Actions and images are generated in parallel within the same denoising step. Given the quantized future image tokens in a fixed-length sequence $\mathbf{v}_0 = (v_{0,1}, \ldots, v_{0,j}, \ldots, v_{0,L_v})$ is a $L_v$-dimension vector denoted by $v_{0,j} \in \{1, ..., V_v\}$ and the variable-length quantized action tokens $\mathbf{a}_0 = (a_{0,1}, \ldots, a_{0,i}, \ldots, a_{0,L_a})$ is a $L_a$-dimension vector denoted by $a_{0,i} \in \{1, ..., V_a\}$, the complete sequence in JD3P is:

$$\mathbf{v}_0, \mathbf{a}_0 = (v_{0,1}, \ldots, v_{0,L_v}, a_{0,1}, \ldots, a_{0,L_a}). \tag{2}$$

We augment the vocabulary with a special mask token M (*i.e.*, `<MASK>`), yielding $V_m = V_v + V_a + 1$ symbols and one-hot basis $\{\mathbf{e}_1, \ldots, \mathbf{e}_{V_v+V_a}, \mathbf{e}_M\}$.

The **noising** process of JD3P is a Markov chain $\{\mathbf{v}_t, \mathbf{a}_t\}_{t=0}^T$ with per-step transition matrices $\mathbf{Q}_t \in \mathbb{R}^{V_m \times V_m}$ that independently map each token to M with probability $\beta_t$ and keep it unchanged with probability $1-\beta_t$. Formally, for any one-hot vector $\mathbf{e}_{t,r}$ of token $v_{t,j}$ and $a_{t,i}$,

$$\mathbf{Q}_t \, \mathbf{e}_{t,r} = (1-\beta_t) \, \mathbf{e}_{t,r} + \beta_t \, \mathbf{e}_M, \tag{3}$$

where $r$ denotes any position in the concatenated sequence $\mathbf{v}_t, \mathbf{a}_t$. The transition matrices compose as $\bar{\mathbf{Q}}_t = \mathbf{Q}_t \cdots \mathbf{Q}_1$, leading to a corrupted distribution at time $t$ that factorizes position-wise:

$$q(\mathbf{v}_t, \mathbf{a}_t \mid \mathbf{v}_0, \mathbf{a}_0) = \prod_{r=1}^{L_v+L_a} \mathrm{C}\left(v_{t,j}, a_{t,i} \mid \bar{\mathbf{Q}}_t \mathbf{e}_{0,r}\right), \tag{4}$$

where C denotes categorical distribution.

The **denoising** process defines conditionals $p_\theta(\mathbf{v}_{t-1}, \mathbf{a}_{t-1} \mid \mathbf{v}_t, \mathbf{a}_t, \mathbf{c})$ under input context $\mathbf{c}$ (*i.e.*, text and current image) as

$$p_\theta(\mathbf{v}_{t-1}, \mathbf{a}_{t-1} \mid \mathbf{v}_t, \mathbf{a}_t, \mathbf{c}) = p_\theta(\mathbf{v}_{t-1} \mid \mathbf{v}_t, \mathbf{c}) \, p_\theta(\mathbf{a}_{t-1} \mid \mathbf{v}_t, \mathbf{a}_t, \mathbf{c}). \tag{5}$$

Considering the mask, it can be decomposed as

$$p_\theta(v_{t-1,j} \mid \mathbf{v}_t, \mathbf{c}) = \left[\delta(v_{t-1,j} = v_{t,j})\right]^{1-\mathbb{1}\{v_{t,j}=M\}} \left[\mathrm{C}\left(v_{t-1,j} \mid \pi_\theta^{(v)}(j \mid \mathbf{v}_t, \mathbf{c})\right)\right]^{\mathbb{1}\{v_{t,j}=M\}} \tag{6}$$

and

$$p_\theta(a_{t-1,i} \mid \mathbf{v}_t, \mathbf{a}_t, \mathbf{c}) = \left[\delta(a_{t-1,i} = a_{t,i})\right]^{1-\mathbb{1}\{a_{t,i}=\mathrm{M}\}} \left[\mathrm{C}\big(a_{t-1,i} \mid \pi_\theta^{(a)}(i \mid \mathbf{v}_t, \mathbf{a}_t, \mathbf{c})\big)\right]^{\mathbb{1}\{a_{t,i}=\mathrm{M}\}}. \tag{7}$$

Here, $\mathbb{1}$ is the indicator function (1 if the condition is true, 0 otherwise), $\delta(\cdot)$ is the Kronecker delta (1 if the arguments are equal, 0 otherwise), $\pi_\theta^{(v)}(j \mid \cdot)$ and $\pi_\theta^{(a)}(i \mid \cdot)$ correspond to the model's predictive distributions over visual and action tokens. At each denoising step, UD-VLA selectively reconstructs a subset of masked positions while leaving clean others hidden, thereby annealing the mask ratio from high to low until the original signals $\mathbf{v}_0$ and $\mathbf{a}_0$ are recovered.

**Loss Function.** We discard the explicit diffusion chain and adopt a single–step *mask-predict* objective. At each update we sample a mask ratio $\rho_t \in (0, 1]$ and apply it to the clean sequences $\mathbf{v}_0$ and $\mathbf{a}_0$ by replacing the selected positions with `<MASK>`, yielding $\mathbf{v}_t$ and $\mathbf{a}_t$. We then optimize the VLA $p_\theta$ by maximum likelihood restricted to the masked sites, computing cross-entropy only on those positions to recover the original tokens.

$$\mathcal{L}_{\mathrm{CE}}(\theta) = -\omega \sum_j^{L_v} \log p_\theta^{(v)}\big(v_{0,j} \mid \mathbf{v}_t, \mathbf{c}\big) \cdot \mathbb{1}\{v_{t,j} = \mathrm{M}\} - \sum_i^{L_a} \log p_\theta^{(a)}\big(a_{0,i} \mid \mathbf{v}_t, \mathbf{a}_t, \mathbf{c}\big) \cdot \mathbb{1}\{a_{t,i} = \mathrm{M}\}, \tag{8}$$

where $\omega$ down-weights the visual tokens to avoid their dominance. This design promotes stronger vision–action interaction and replaces the multi-step corruption chain with a single-step masking objective.

## 3.2 TRAINING

We initialize our model from a pretrained VLM backbone (Wang et al., 2024), which is trained in an autoregressive manner with causal attention. At the beginning, we adopt a stage (i) to post-train UD-VLA on a large-scale video dataset, where the token sequences are constructed as:

$$[\text{ text tokens } ; \text{ current image tokens } ; \text{ future image tokens}]. \tag{9}$$

This stage injects capabilities of future image generation into the VLA, enabling the robot to understand and model future states. In stage (ii), we jointly optimize image and action generation under a unified framework on the downstream robot action dataset. The token sequences are constructed as Equation (1). Specifically, we reformulate the autoregressive decoding as a diffusion process following JD3P. Unlike standard diffusion models that predict tokens at masked positions, we adopt a shift operation strategy to predict the next token, as introduced in Gong et al. (2025). This approach allows the model to retain the capacity learned from next-token prediction training, while also benefiting from bidirectional context and parallel decoding.

## 3.3 INFERENCE

At inference, the denoising process is instantiated through parallel decoding with adaptive masking: (i) all positions of $\mathbf{v}_T$ and $\mathbf{a}_T$ are initialized as `<MASK>`, (ii) token distributions for all positions are predicted in parallel, and (iii) this procedure is repeated for a small number of iterations.

**Prefix KV Cache and Pre-filling Tokens.** Our UD-VLA employs prefix KV-Cache to cache the keys and values of current visual and prompt tokens. Besides, because the length of visual tokens is fixed while action tokens have varied length, we pre-fill the `<BOI>`, `<EOI>`, and `<BOA>` tokens at the corresponding positions, which guides the denosing of visual tokens and action tokens. Our experiments empirically show that the cache and pre-filling reduce latency and contribute to higher speed.

**Confidence-Guided Decoding.** We initialize sequence with noise at $t = T$ and iterate backward to $t = 0$, applying the cosine mask schedule $\rho_t = \cos\left(\frac{\pi}{2}\frac{T+1-t}{T+1}\right)$ for $t = T, \dots, 1$, which yields smoother sampling and refinement. Follow Equation (5), the model evaluates the joint reverse conditionals at each step. We rank only the currently masked positions and let $M_t \subseteq \{1, \dots, L_v + L_a\}$ be the masked set at step $t$ with size $|M_t|$. For each location $r$, a confidence score is computed:

$$q_{t-1,r} = \max_\ell \begin{cases} p_\theta\big(\ell \mid \mathbf{v}_t, \mathbf{u}\big), & r \in \{1, \dots, L_v\}, \\ p_\theta\big(\ell \mid \mathbf{v}_t, \mathbf{a}_t, \mathbf{u}\big), & r \in \{L_v + 1, \dots, L_v + L_a\}. \end{cases} \tag{10}$$

We then update the top $(1 - \rho_t)|M_t|$ entries among the masked indices:

$$\Omega_t = \operatorname*{TopK}_{(1-\rho_t)|M_t|} \left\{ q_{t-1,r} : r \in M_t \right\}. \tag{11}$$

For visual indices $j$ and action indices $i$ in $\Omega_t$, tokens are updated via tempered Gumbel sampling:

$$v_{t-1,j}, a_{t-1,i} = \operatorname*{GumbelMax}_y p_\theta\big(y \mid \mathbf{v}_t, \mathbf{a}_t, \mathbf{u}\big), \qquad (j,i) \in \Omega_t, \tag{12}$$

where $\mathrm{GumbelMax}$ denotes sampling using the Gumbel-max trick.

**Decoding Space Mapping.** The tokenized image and action tokens come from small codebooks that constitute only a subset of the model's vocabulary. During inference, we restrict the classification space within each modality to its designated range. This prevents the model from producing tokens of the wrong modality and stabilizes the generation process. In addition, once an `<EOA>` is predicted at index $i^\star$, we fix the action length and deterministically replace all subsequent tokens $a_{t,i_{i>i^\star}}$ with `<MASK>`, preventing them from corrupting action prediction.

## 4 EXPERIMENTS

To comprehensively evaluate the effectiveness of our UD-VLA, we conduct experiments on three simulated benchmarks as well as a real-world robotic platform, comparing against widely adopted baselines and ablated methods. Section 4.1 first introduces all benchmarks. Then, Section 4.2 presents results on simulated benchmarks, where our method achieves SOTA performance across diverse paradigms. Third, we provide an in-depth analysis to assess the contributions of each component in Section 4.3. At last, real-world experiments demonstrate the strong performance and generalization capability of our UD-VLA in Section 4.4. All used baselines are detailed in Section G.

### 4.1 BENCHMARKS

**CALVIN.** The CALVIN benchmark (Mees et al., 2022) is a simulated suite for evaluating long-horizon, language-conditioned robotic manipulation. It spans four environments (A, B, C, and D) with 34 tasks and 1,000 language instructions. We evaluate 500 rollouts per model, where each rollout involves a sequence of 5 consecutive sub-tasks. We report the average length (avg. len.) of successful sub-task completions of all rollouts with a maximum value of 5.

**LIBERO.** LIBERO (Liu et al., 2023) is a simulated manipulation benchmark with 4 suites (Spatial, Object, Goal, Long). Spatial probes layout reasoning, Object tests object generalization, Goal evaluates goal-conditioned control, and Long targets long-horizon compositional skills. We report success rates per suite and overall average, each suite containing 10 tasks and 50 rollouts per task.

**SimplerEnv.** SimplerEnv (Li et al., 2024b) is a real-to-sim suite for assessing transfer and generalization of robot policies trained on real-world video data. We evaluate on WidowX robots under varied lighting, textures, colors, and viewpoints. Tasks include *Put Spoon on Towel*, *Put Carrot on Plate*, *Stack Green on Yellow Block*, and *Put Eggplant in Yellow Basket*. We report per-task success rates and the overall average.

### 4.2 MAIN RESULTS IN SIMULATION

**CALVIN.** Table 2 shows that UD-VLA achieves an average success length of 4.64 on CALVIN ABCD→D benchmark, outperforming all baselines and demonstrating strong capabilities in long-horizon reasoning and execution. Compared with UniVLA, which autoregressively predicts future frames during post-training while only inferring actions, our superior performance suggests that explicit visual generation serves as a form of chain-of-thought (CoT), providing more effective guidance for action prediction. In contrast to UP-VLA, which predicts both future frames and actions in a single step on filled tokens, our multi-step diffusion more efficiently facilitates information exchange between generated images and actions.

**LIBERO.** Table 3 shows that UD-VLA achieves an average success rate of 96.1%, which is SOTA performance on the LIBERO benchmark. In particular, it attains 98.8% on the Object suite and 95.2%

Table 2: **Comprehensive Evaluation of Long-Horizon Robotic Manipulation on the CALVIN Benchmark.** UniVLA* denotes the variant without historical frames for fair comparison.

| Method | Task | Tasks Completed in a Row | | | | | Avg. Len ↑ |
|---|---|---|---|---|---|---|---|
| | | 1 | 2 | 3 | 4 | 5 | |
| MCIL (Lynch & Sermanet, 2020) | ABCD→D | 0.373 | 0.027 | 0.002 | 0.000 | 0.000 | 0.40 |
| RT-1 (Brohan et al., 2023) | ABCD→D | 0.844 | 0.617 | 0.438 | 0.323 | 0.227 | 2.45 |
| Robo-Flamingo (Li et al., 2024a) | ABCD→D | 0.964 | 0.896 | 0.824 | 0.740 | 0.660 | 4.09 |
| GR-1 (Wu et al., 2024) | ABCD→D | 0.949 | 0.896 | 0.844 | 0.789 | 0.731 | 4.21 |
| ReconVLA (Song et al., 2025c) | ABCD→D | 0.980 | 0.900 | 0.845 | 0.785 | 0.705 | 4.23 |
| UniVLA* (Wang et al., 2025) | ABCD→D | 0.948 | 0.906 | 0.862 | 0.834 | 0.690 | 4.26 |
| MODE (Reuss et al., 2025) | ABCD→D | 0.971 | 0.925 | 0.879 | 0.835 | 0.779 | 4.39 |
| UP-VLA (Zhang et al., 2025a) | ABCD→D | 0.962 | 0.921 | 0.879 | 0.842 | 0.812 | 4.42 |
| MDT (Reuss et al., 2024) | ABCD→D | 0.986 | 0.958 | 0.916 | 0.862 | 0.801 | 4.52 |
| **UD-VLA** | ABCD→D | **0.992** | **0.968** | **0.936** | **0.904** | **0.840** | **4.64** |

Table 3: **Evaluation and comparison on the LIBERO benchmark.**

| Method | Spatial | Object | Goal | Long | Average |
|---|---|---|---|---|---|
| Octo (Octo Team et al., 2024) | 78.9% | 85.7% | 84.6% | 51.1% | 75.1% |
| SpatialVLA (Qu et al., 2025) | 88.2% | 89.9% | 78.6% | 55.5% | 78.1% |
| CoT-VLA (Zhao et al., 2025) | 87.5% | 91.6% | 87.6% | 69.0% | 81.1% |
| $\pi_0$-FAST (Pertsch et al., 2025) | 96.4% | 96.8% | 88.6% | 60.2% | 85.5% |
| WorldVLA (Cen et al., 2025) | 87.6% | 96.2% | 83.4% | 60.0% | 81.8% |
| FlowVLA (Zhong et al., 2025) | 93.2% | 95.0% | 91.6% | 72.6% | 88.1% |
| DreamVLA (Zhang et al., 2025b) | 97.5% | 94.0% | 89.5% | 89.5% | 92.6% |
| OpenVLA-OFT (Kim et al., 2025) | 96.2% | 98.3% | 96.2% | 90.7% | 95.3% |
| UniVLA (Wang et al., 2025) | 95.4% | 98.8% | 93.6% | 94.0% | 95.5% |
| F1 (Lv et al., 2025) | **98.2%** | 97.8% | **95.4%** | 91.3% | 95.7% |
| **UD-VLA** | 96.2% | **98.8%** | 94.2% | **95.2%** | **96.1%** |

on the Long suite, demonstrating robust generalization in object recognition and strong temporal reasoning capabilities through future-frame prediction. Our method outperforms approaches that unify modalities through extrinsic experts (e.g., DreamVLA, F1) as well as those that unify input and output spaces (e.g., CoT-VLA, UniVLA, WorldVLA), demonstrating the advantage of our end-to-end unified VLA paradigm with joint denoising. Compared to F1, which builds a progressive generation process for future images and actions, UD-VLA delivers higher performance, highlighting that the joint discrete diffusion denoising process enables more effective fusion of the two modalities.

**SimplerEnv.** On the SimplerEnv-WidowX benchmark (see Table 4), UD-VLA achieves an average success rate of 75.0%, significantly outperforming all baselines. Unlike methods (Qu et al., 2025), that require additional inputs, UD-VLA predicts precise motion and grasp targets by explicitly modeling future frames. Our method outperforms F1, demonstrating its potential as a superior paradigm for integrating understanding, generation, and acting. We also surpass UniVLA, which requires additional historical information as input, highlighting the advantages of incorporating visual information into the reasoning process. In the stack block task, which requires precise manipulation, our approach achieves a 37.5% higher success rate than SpatialVLA with 3D perception capabilities.

### 4.3 IN-DEPTH ANALYSIS

**Effectiveness of Hybrid Attention Mechanism.** As shown in Table 5, the hybrid attention design achieves better performance than purely bidirectional or causal attention mechanism on CALVIN, reaching an average success length of 4.64. This demonstrates that hybrid attention achieves the most effective information transfusion. Tokens in one image are able to attend to each other to ensure better global consistency. Likewise, actions of different dimensions, such as position and rotation, do not follow a strict causal

Table 5: **Effectiveness of different attention schemes.** *Bidirectional* applies bidirectional attention over the visual generation and the action block. *Causal* uses a strict lower-triangular mask, and *Hybrid* follows Figure 2.

| Attention | Avg. Len. |
|---|---|
| Causal | 4.04 |
| Bidirectional | 4.32 |
| **Hybrid (ours)** | **4.64** |

Table 4: **Evaluation results on SimplerEnv-WidowX .**

| Model | Put Spoon | Put Carrot | Stack Block | Put Eggplant | Overall |
|---|---|---|---|---|---|
| OpenVLA (Kim et al., 2025) | 0.0% | 0.0% | 0.0% | 4.1% | 1.0% |
| Octo (Octo Team et al., 2024) | 47.2% | 9.7% | 4.2% | 56.9% | 29.5% |
| RoboVLMs (Liu et al., 2025) | 45.8% | 20.8% | 4.2% | 79.2% | 37.5% |
| SpatialVLA (Qu et al., 2025) | 16.7% | 25.0% | 29.2% | **100%** | 42.7% |
| F1 (Lv et al., 2025) | 50.0% | **70.8%** | 50.0% | 66.7% | 59.4% |
| Univla (Wang et al., 2025) | **83.3%** | 66.7% | 33.3% | 95.8% | 69.8% |
| **UD-VLA** | **83.3%** | 62.5% | **66.7%** | 91.6% | **76.0%** |

Table 6: **Target of Visual Generation.** *Null* denotes action prediction without visual generation. *Current Image* denotes reconstruct current observation.

| Target | Avg. Len. |
|---|---|
| Null | 4.21 |
| Current Image | 4.39 (*+0.18*) |
| **Future Image (ours)** | **4.64** (*+0.43*) |

Table 7: **Decoding Mechanism.** *AR* denotes autoregressive decoding, *Jacobi* denotes the parallel decoding scheme used in (Song et al., 2025b), and *ID* denotes independent diffusion of future images and actions.

| Method | Avg. Len. | Speed (tokens/s) |
|---|---|---|
| AR | 4.18 | 50.2 (*×1.0*) |
| Jacobi | 4.16 (*-0.02*) | 101.6 (*×2.0*) |
| ID | 4.35 (*+0.19*) | 144.4 (*×2.9*) |
| **JD3P (ours)** | **4.64** (*+0.46*) | **219.3** (*×4.3*) |

order, and bidirectional attention enables the model to capture their correlations more effectively. However, bidirectional attention across modalities leads to information leakage, as mentioned in Section 3.1, demonstrating 0.3 lower average length.

**Effectiveness of Future Image Generation.** In Table 6, we compare three settings: not generating visual information, reconstructing current frames, and predicting future frames. While reconstructing the current images improves the model's fine-grained perception and thereby strengthens action prediction, it remains limited because the model only learns static scene information. In contrast, jointly predicting future images and actions provides richer temporal cues, allowing the model to anticipate visual dynamics and align them with action planning, which leads to the best performance.

**Effectiveness of JD3P.** As shown in Table 7, autoregressive decoding is not well-suited for visual generation, since image tokens are poorly modeled by next-token generation and the process is extremely slow with only 50.2 tokens per second and a limited average length of 4.18. The Jacobi method improves decoding speed $2\times$, but its performance is still constrained by the inherent limitations of the autoregressive model. The independent diffusion process allows the optimization of visual generation and action prediction iteratively, which naturally suits them (Deng et al., 2025; Black et al., 2024), thus reaching an average length of 4.35 and a much higher speed. In contrast, JD3P attains both the highest success rate of 4.64 and the $4.3\times$ faster decoding speed, demonstrating that it is a more effective and efficient approach for unifying visual-action generation. This demonstrates that joint denoising enables the actions to better refine their predictions by leveraging image information from intermediate denoising steps, which can be regarded as a computational scaling. In contrast, the independent diffusion process allows for only limited information flow, because it predicts actions with only a single computation conditioned on future images.

## 4.4 REAL-WORLD EXPERIMENT

**Setup and Tasks.** The real-world setup is shown in Figure 3, with a detailed description provided in Section C. We collect three categories of tasks, namely *stacking bowls*, *putting blocks (into a box)*, and *flipping towers*. Each category includes objects with varied colors and shapes, and data is captured under three distinct backgrounds. For each category, we record 200 trajectories at 15 Hz. During evaluation, we consider both seen and unseen settings, where unseen tasks feature novel scenes and objects.

**Results and Analysis.** We evaluate each methods for 30 times. Figure 3 shows that our UD-VLA consistently outperforms the GR00T N1 (Bjorck et al., 2025) and UniVLA (Wang et al., 2025) baselines across all tasks, achieving success rates above 80%. For **seen tasks**, our action quantization brings the precision of action representations, while our joint denoising process ensures the quality

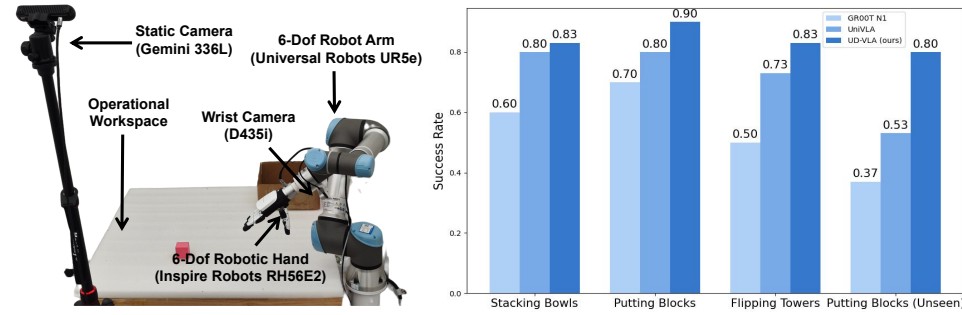

Figure 3: **Real-world Setup and Results.**

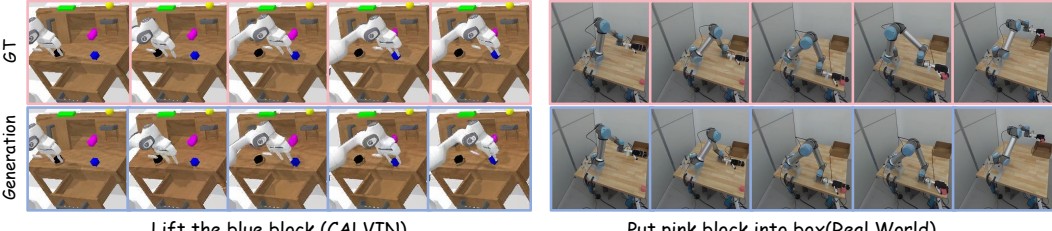

Lift the blue block (CALVIN)                    Put pink block into box(Real World)

Figure 4: **Visualization of Generated Future Frames and Ground-truth Frames.** The leftmost frame of each trajectory represents the current observation.

of the actions. For **unseen tasks**, GR00T N1 exhibits poor generalization on unseen targets and backgrounds, while our UD-VLA owns visual generalization to generate images with unseen targets, which further leads to correct actions. Although UniVLA also benefits from visual post-training, its inference lacks explicit visual reasoning, thus falling short on unseen objects. In contrast, our joint modeling of future visual tokens and actions yields markedly stronger generalization to unseen scenes, leading to higher success rates. In addition, visualizations can be found in Section B.

### 4.5 Visualization of Future Image Generation

To intuitively illustrate the effectiveness and limitations of image generation, we present representative examples of generated future images across simulation and the real world on different embodiments in Figure 4. Overall, the predicted frames follow instructions closely and capture task-level dynamics, remaining well aligned with the ground-truth trajectories. This consistency indicates that the model has internalized a temporal understanding of task logic, which in turn enhances temporal reasoning and enables more coherent action generation. However, the generation lacks visual fidelity, especially in fine-grained details such as robotic arms and backgrounds. This limitation arises from the absence of large-scale generative pretraining and the use of compressed images with few tokens for efficiency. Despite these shortcomings, the generated frames reliably convey task progression and remain informative for downstream control. While pixel-level accuracy is difficult, the model consistently produces foresight images sufficient for action planning.

### 5 Conclusion

We propose a unified diffusion VLA that unifies understanding, generation, and acting through a joint discrete denoising diffusion process. It jointly refines image and action tokens via a single synchronous denoising trajectory. We construct a unified multimodal space and a hybrid attention mechanism as foundations. Besides, we design a two-stage training and several test-time techniques to balance performance and efficiency. Experiments show that our method achieves state-of-the-art results in both simulation and real-world environments.

## ACKNOWLEDGMENT

This work was supported in part by the Natural Science Foundation of China under Grant 62403401, in part by the Guangdong Basic and Applied Basic Research Foundation under Grant 2024A1515011992, in part by the Guangdong Provincial Project under Grant 2024QN11X127, and in part by the AI Research and Learning Base of Urban Culture under Grant 2023WZJD008.

## REPRODUCIBILITY STATEMENT

The paper and appendix detail the full experimental setup, including data, action/image tokenization, model architectures, training schedules, action chunk lengths, decoding methods, and selection criteria (see Section 4, Section C and Section F). Evaluation protocols are specified for both benchmarks, strictly following the official splits. All datasets used are publicly available, enabling consistent and verifiable evaluation. These details are provided to enable re-implementation and verification of our results.

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

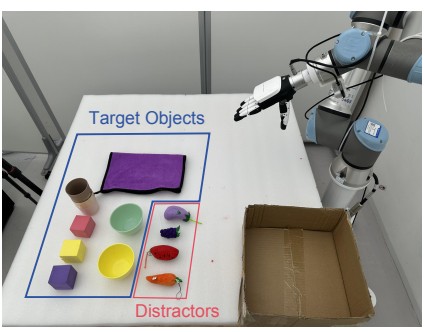 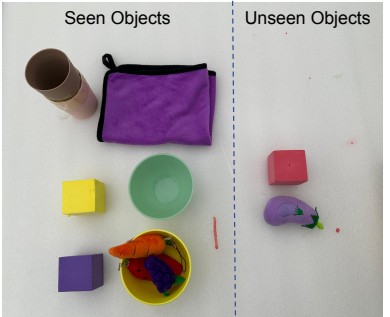 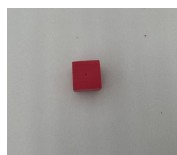 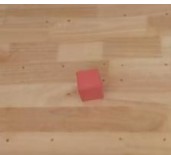

Figure 5: **Details of our real-world experiment settings. Left:** The overview of the systems, including target objects that the robot is expected to interact with, and distractors that are used to interfere with the model's actions. **Middle:** The objects are divided into seen ones that exist in the training sets and unseen ones that are used for the unseen evaluation of "putting blocks". **Right:** The backgrounds are divided into seen and unseen, where the unseen background is used for the unseen evaluation of "putting blocks".

## A  LLM USAGE STATEMENT

We used large language models only as general writing aids for light proofreading and language refinement in grammar, phrasing, and minor style. The research ideas, problem formulation, methodology, experiments, analyses, code, and substantive writing were conceived and authored by us. No large language model generated new content beyond localized edits, and none was used to design experiments, analyze data, write code, or draft sections. All text was reviewed and verified by the authors, who take full responsibility for the paper, and the large language model should not be regarded as a contributor.

## B  VISUALIZATION OF FUTURE IMAGE GENERATION

To intuitively illustrate the effectiveness and limitations of image generation, we present representative examples of generated future images across simulation and the real world on different embodiments in Figure 4. Overall, the predicted frames follow instructions closely and capture task-level dynamics, remaining well aligned with the ground-truth trajectories. This consistency indicates that the model has internalized a temporal understanding of task logic, which in turn enhances temporal reasoning and enables more coherent action generation. However, the generation lacks visual fidelity, especially in fine-grained details such as robotic arms and backgrounds. This limitation arises from the absence of large-scale generative pretraining and the use of compressed images with few tokens for efficiency. Despite these shortcomings, the generated frames reliably convey task progression and remain informative for downstream control. While pixel-level accuracy is difficult, the model consistently produces foresight images sufficient for action planning.

## C  REAL-WORLD SETUP

Our real-world setup consists of a 6-DoF UR5e robotic arm equipped with a 6-DoF Inspire RH56E2 robotic hand for dexterous manipulation. A wrist-mounted Intel RealSense D435i depth camera provides close-range RGB-D observations of the manipulated objects, while a static Gemini 336L camera captures the entire operational workspace from a fixed viewpoint.

In Figure 5, we first overview our systems, including target objects that the robot is expected to interact with, and distractors. Then, we illustrate the unseen settings in detail, including unseen objects and an unseen background that do not exist in the training set. The unseen settings pose a challenge for the visual generalization of the models. In this case, our model can generate reliable future images as visual plans and further guide precise action generation, thus outperforming baselines.

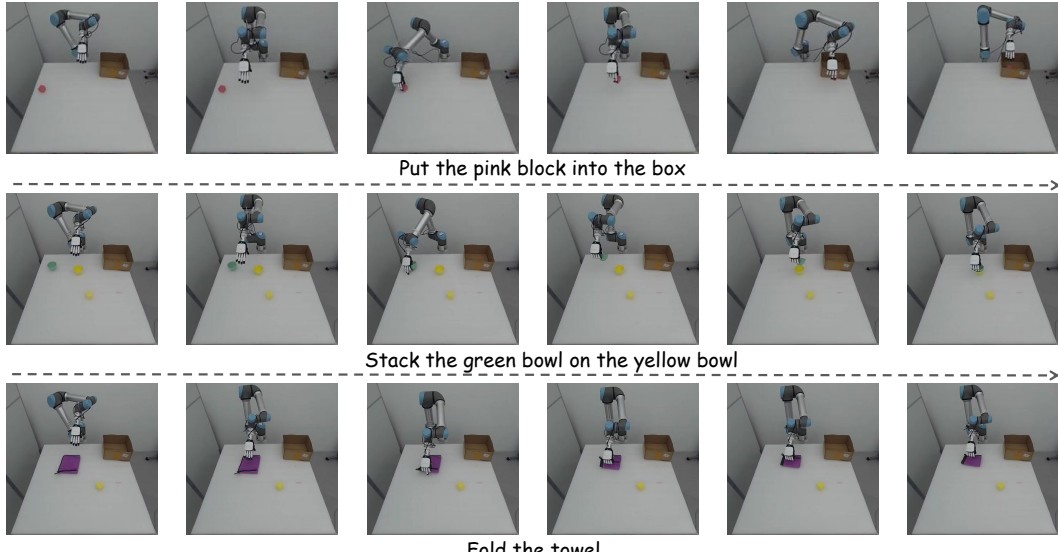

Figure 6: Visualization of our tasks. Each row shows a representative trajectory of one task.

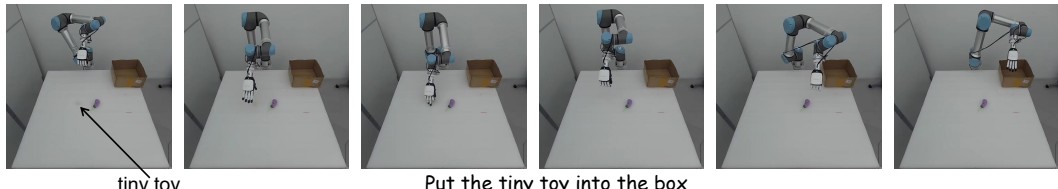

Figure 7: Visualization of tasks that require precise manipulation. The tiny toy used in these experiments has a radius of approximately 1.5 cm, which is about one–fifth the size of the other objects.

## D    REAL-WORLD TASKS

**Task Definition.** We carefully introduce our three real-world tasks and provide visualizations in Figure 6:

- Stacking Bowls: The robot is expected to lift one bowl and stack it on the other.
- Putting Blocks: The robot is expected to lift the target block and put it into the box.
- Flipping Towers: The robot is expected to grasp one side of the tower and move to the other side, until the towels are folded.

**Training Details.** We train our model for real-world experiments on 4 NVIDIA H100 GPUs for 24 hours, a total of 9000 steps. The batch size is 64, chunk size is 8, and the learning rate is 8e-5 with weight decay of 0.1.

## E    LOSS FORMULATIONS

**Pixel-level Mean Squared Error.** In Table 1, $\mathcal{L}_{\mathrm{MSE}}$ denotes a reconstruction objective that measures the average squared discrepancy between the predicted output and the ground-truth target, applicable to both visual and action modalities. The loss is defined as

$$\mathcal{L}_{\mathrm{MSE}} = \frac{1}{|\Omega|} \sum_{u \in \Omega} \left\| x(u) - \hat{x}_\theta(u \mid c) \right\|_2^2, \tag{13}$$

where $x$ is the ground-truth target, $\hat{x}_\theta(\cdot \mid c)$ is the model prediction conditioned on $c$ (e.g., instruction, proprioception), $\Omega$ is the index set over which the discrepancy is computed, and $u$ enumerates elements in $\Omega$.

**Continuous Diffusion.** In Table 1, $\mathcal{L}_{\text{Diff-cont}}$ denotes a conditional diffusion objective that learns to predict the injected noise for a noised sample at timestep $t$ (optionally conditioned on $c$), enabling iterative denoising back to the clean data. Formally,

$$\mathcal{L}_{\text{Diff-cont}} = \mathbb{E}_{t,\, x_0 \sim q(x_0),\, \epsilon \sim \mathcal{N}(0,I),\, c} \big\| \epsilon - \epsilon_\theta(x_t, t, c) \big\|_2^p, \quad x_t = \alpha_t x_0 + \sigma_t \epsilon, \ \ p \in \{1, 2\}, \quad (14)$$

where $x_0$ is the clean continuous target, $x_t$ is its noised version at timestep $t$, $\epsilon$ is Gaussian noise, $c$ is an optional conditioning signal (e.g., instruction, proprioception), $\alpha_t$ and $\sigma_t$ are the noise schedule coefficients at timestep $t$, $t$ is the diffusion timestep sampled from a predefined schedule, and $p \in \{1, 2\}$ denotes the norm used for the loss (typically $p = 2$ for L2 or $p = 1$ for L1/Smooth-L1). The model predicts $\epsilon$ and is then used in reverse updates $x_{t-1} = F_\theta(x_t, t, c)$ to reconstruct $x_0$.

By default $\mathcal{L}_{\text{Diff-cont}}$ uses an L2 error for noise prediction. Some implementations (e.g., GR-1 (Wu et al., 2024), SEER (Tian et al., 2025)) replace it with Smooth-L1 (Huber) without altering the diffusion formulation. Formally, replace the L2 error with Smooth-L1

$$\rho_\beta(r) = \begin{cases} \dfrac{1}{2} \dfrac{\|r\|_2^2}{\beta}, & \|r\|_2 < \beta, \\[2mm] \|r\|_2 - \dfrac{1}{2}\beta, & \text{otherwise,} \end{cases} \quad (15)$$

with $r = \epsilon - \epsilon_\theta(x_t, t, c)$ and $\beta > 0$ (default $\beta = 1$ unless specified). The diffusion objective becomes $\mathbb{E}[\rho_\beta(r)]$.

**Discrete Diffusion.** In Table 1, $\mathcal{L}_{\text{Diff-disc}}$ denotes a masked-token prediction objective for discrete sequences over a finite vocabulary (including a special `[MASK]`). Formally,

$$\mathcal{L}_{\text{Diff-disc}} = \mathbb{E}_M \left[ -\frac{1}{|M|} \sum_{i \in M} \log p_\theta\big(z_i^* \mid z_{\text{masked}}^{(M)}, c\big) \right], \quad (16)$$

where $M$ is the set of masked positions, $z_i^*$ is the ground-truth token at position $i$, $z_{\text{masked}}^{(M)}$ is the visible context obtained by replacing positions in $M$ with `[MASK]`, and $c$ is an optional conditioning signal. The model outputs a categorical distribution over the vocabulary and learns to recover the masked tokens from the visible context.

**Next-Token Prediction.** In Table 1, $\mathcal{L}_{\text{NTP}}$ denotes the standard autoregressive objective (negative log-likelihood over the vocabulary) that maximizes the probability of the next token given its history:

$$\mathcal{L}_{\text{NTP}} = \mathbb{E}_i \big[ -\log P_\theta(x_i \mid x_{<i}) \big], \quad (17)$$

where $x_{<i}$ is the preceding context and $x_i$ is the next token. This loss encourages the model to assign high probability to the ground-truth continuation under a causal (history-only) factorization.

## F  TRAINING DETAILS.

The training for the three virtual environment experiments and the real-world experiments was conducted on H100 GPUs. For the Calvin-ABCD dataset, we set the action chunk to 10 and trained for approximately 24 hours across 8 H100 GPUs. For the LIBERO dataset, we jointly trained on four tasks instead of training each task separately, with the action chunk set to 10, and the training lasted around 30 hours on 8 H100 GPUs. In the SimplerEnv environment, we used the Bridge dataset for training, with the action chunk set to 5, and trained for about 30 hours on 8 H100 GPUs. For the real-world experiments, we collected over 600 trajectories across 3 tasks, designed with an action chunk of 8, and trained for around 8 hours on 8 H100 GPUs.

## G  BASELINES

**MCIL (Lynch & Sermanet, 2020).** MCIL introduces language-conditioned robotic manipulation with a single end-to-end visuomotor policy that learns perception from pixels, natural language understanding, and multitask continuous control. A key contribution is multicontext imitation learning,

which enables training from largely unstructured and unlabeled demonstrations (no task or language labels), reducing language annotation to under 1% of the dataset while improving language-conditioned performance. At test time, the policy follows long-horizon free-form instructions in a 3D tabletop environment using only text prompts. MCIL further combines any language-conditioned policy with large pretrained language models to handle many out-of-distribution synonym instructions across multiple languages without collecting new demonstrations.

**RT-1 (Brohan et al., 2023).** RT-1 is a single Transformer visuomotor policy that encodes camera images, natural language instructions, and motor commands into token sequences for real-time control. It is trained on over 130k demonstrations collected across 13 robots. RT-1 reports 97% success on over 700 instructions, improved generalization to novel tasks, distractors, and backgrounds. RT-1 can also execute long-horizon procedures within SayCan (Ahn et al., 2022), with as many as 50 stages. RT-1 can also absorb heterogeneous data from simulation and other robot morphologies while retaining performance on the original tasks.

**Robo-Flamingo (Li et al., 2024a).** RoboFlamingo builds on the open-source VLM OpenFlamingo to form a vision–language manipulation framework. It decouples visual–language understanding and decision making: the pretrained VLM is used for per-step comprehension of images and instructions, while an explicit policy head models history. Then the policy is fine-tuned solely on language-conditioned manipulation datasets via imitation learning. This design lets a small amount of demonstrations adapt the model to downstream tasks, supports open-loop control. On the CALVIN benchmark, it outperforms prior baselines (Brohan et al., 2023) and demonstrates zero-shot generalization to new settings.

**GR-1 (Wu et al., 2024).** GR-1 is a straightforward GPT-style transformer for multi-task, language-conditioned visual robot manipulation. It takes as input a language instruction, a sequence of observation images, and robot states, and predicts both robot actions and future images end-to-end. The model is first pre-trained for video prediction on a large-scale video dataset, then seamlessly fine-tuned on robot data. On the CALVIN benchmark, the paper reports improvements over contemporaneous baselines (e.g., success rate from 88.9% to 94.9%, average length from 3.06 to 4.21), strong zero-shot unseen-scene generalization (53.3% to 85.4%). Real-robot experiments further show performance gains and robustness to unseen scenes and objects. **OpenVLA (Kim et al., 2025).** OpenVLA is a 7B-parameter open-source VLA trained on 970k real-world robot manipulation trajectories from Open-X Embodiment. It uses a visually conditioned Llama-2 backbone and fuses pretrained features from DINOv2 and SigLIP, capturing visual cues at multiple granularities. The paper reports strong generalist manipulation results, outperforming the 55B RT-2-X by 16.5% absolute success across 29 tasks and multiple embodiments with roughly 7× fewer parameters.

**ReconVLA (Song et al., 2025c).** ReconVLA is a reconstructive VLA with an implicit grounding paradigm that addresses the observation of dispersed visual attention in prior VLAs. It takes current images, language instructions, and robot proprioception as inputs. By reconstructing gaze regions corresponding to target objects, the policy acquires fine-grained representations and allocates attention to task-relevant regions, enabling precise manipulation. The authors also curate a large-scale pretraining corpus (over 100k trajectories and 2M samples) from open-source robotic datasets to improve generalization in visual reconstruction. Experiments in long-horizon simulation and real-world settings report superior performance, directive attention visualizations, and generalization to unseen objects and scenes.

**UP-VLA (Zhang et al., 2025a).** UP-VLA revisits VLA pre-training by coupling multi-modal understanding with future visual prediction to retain both high-level semantics and low-level spatial cues. Concretely, it co-trains an autoregressive policy with a flexible attention mask on heterogeneous datasets, using vision–language understanding to align semantic features and video-style future prediction to capture fine-grained visual patterns. UP-VLA reports that it understands instructions, predicts future images, and plans actions within one framework, with gains across simulated and real-world manipulation; in particular, it improves CALVIN ABC→D generalization by about 33% over prior methods and shows stronger success on tasks requiring precise spatial control, while maintaining competitive in-distribution multitask and real-unseen performance.

**UniVLA (Wang et al., 2025).** UniVLA learns cross-embodiment VLA policies by planning in a unified, task-centric latent action space extracted from videos. The approach derives latent actions in a DINO feature space to separate task-relevant dynamics from irrelevant visual changes, enabling the

use of heterogeneous, web-scale data (including unlabeled human videos) without action labels. A lightweight decoder (about 10.8M parameters) translates latent actions into executable trajectories, supporting efficient adaptation with limited downstream data. The paper reports strong results across manipulation, navigation, and real-robot evaluations, including improvements over OpenVLA (Kim et al., 2025) with a fraction of the compute (1/20 pretraining) and data (1/10 downstream), as well as gains on LIBERO (+18.5%), navigation (+29.6%), and real-world deployments (+36.7%). Performance scales with data diversity and remains robust under embodiment and viewpoint shifts.

**Octo (Octo Team et al., 2024).** OCTO is an open-source, transformer-based generalist robot policy pretrained on 800k trajectories from the Open X-Embodiment dataset. It maps multimodal input tokens (observations and tasks) to action tokens, supporting instruction via language commands or goal images and handling diverse camera configurations and robot platforms. The model can be adapted to new sensory inputs, action spaces, or morphologies by adding lightweight adapters and fine-tuning on a small target dataset within hours on consumer GPUs. OCTO employs a diffusion action head to model expressive action distributions. Experiments across 9 robots and 4 institutions show strong out-of-the-box multi-robot control for single- and dual-arm manipulation and effective initialization for downstream fine-tuning.

**SpatialVLA (Qu et al., 2025).** SpatialVLA targets 3D spatial understanding for generalist robot policies by aligning spatial representations of observations and actions. It introduces Egocentric 3D (Ego3D) Position Encoding to inject 3D context into visual features in the camera frame (avoiding per-robot calibration), and Adaptive Action Grids to discretize continuous robot actions into data-driven spatial grids, learning spatial action tokens that transfer across embodiments. Pretrained on 1.1M real-world robot episodes, SpatialVLA is applied zero-shot to numerous tasks and shows strong in-domain multi-task generalization and long-horizon trajectory inference in both simulation and real robots.

**CoT-VLA (Zhao et al., 2025).** CoT-VLA introduces visual chain-of-thought for robotic control by first generating subgoal images as explicit intermediate reasoning steps and then conditioning actions on the current observation and the generated subgoal. The system is built on a unified multimodal backbone, trained on Open X-Embodiment and action-less video datasets, and fine-tuned on downstream task demonstrations. A hybrid attention design uses causal next-token prediction for text and image generation and full attention to predict action dimensions jointly. The policy further employs action chunking to output short action sequences per step. Experiments in simulation and the real world show that visual chain-of-thought improves policy performance, with reported gains over prior VLAs (e.g., +17% real-world, +6% simulation), and strong results across multiple robot platforms and tasks.

**$\pi$0-FAST (Pertsch et al., 2025).** $\pi$0-FAST proposes a frequency-space action tokenization scheme that compresses robot actions with a discrete cosine transform followed by byte-pair encoding, reducing inter-token correlation and enabling next-token prediction training of autoregressive VLAs on high-frequency and dexterous data. It improves markedly over per-dimension binning and learned VQ tokenizers in simulation and real-robot studies, and makes large-scale training on DROID feasible with zero-shot evaluation via language prompting. Building on this, $\pi$0-FAST is a universal action tokenizer trained on 1M real trajectories across embodiments, action spaces, and control rates, providing an off-the-shelf tokenizer for VLA training.

**WorldVLA (Cen et al., 2025).** WorldVLA is an autoregressive action–world model that unifies image, text and action understanding and generation within a single LLM framework. It encodes images, language, and actions with separate tokenizers that share one vocabulary, so the world module learns environment dynamics by predicting future visuals from actions, while the action module uses visual tokens to generate subsequent actions—yielding mutual enhancement between world modeling and control. The authors observe performance drops when generating multiple actions autoregressively due to error propagation and limited action generalization; they address this with an action-attention masking strategy that selectively masks prior actions when predicting the current one, improving chunked action generation. On LIBERO, WorldVLA improves grasp success by about 4% over an action-only backbone, reduces FVD for video generation by about 10% over a vanilla world model.

**FlowVLA (Zhong et al., 2025).** FlowVLA instantiates a Visual Chain of Thought for world modeling by decomposing next-frame prediction into a motion–then–appearance process, i.e., $v_t \rightarrow$

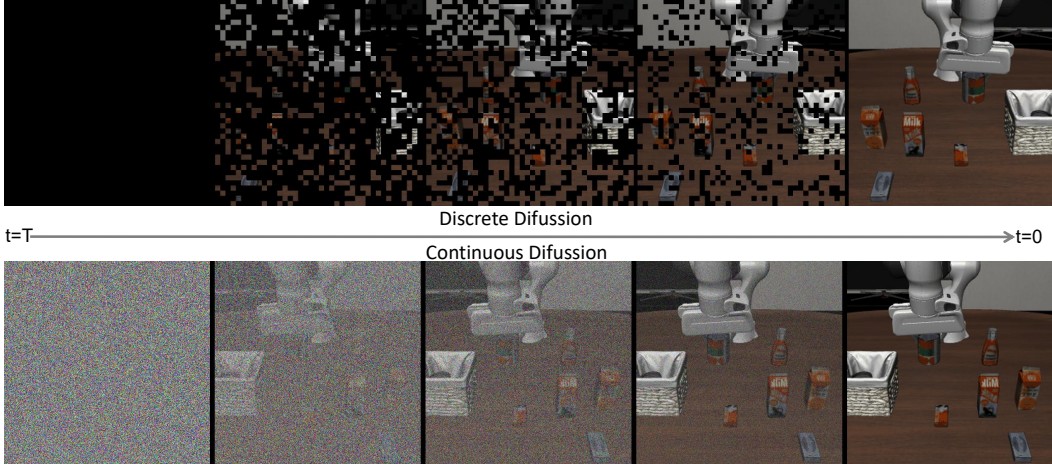

Figure 8: Qualitative Comparison of discrete diffusion and continuous diffusion.

$f_t \to v_{t+1}$ with $f_t$ an intermediate optical-flow target. By first committing to motion, the model learns disentangled dynamics that yield more coherent visual forecasts and more efficient policy learning. Practically, optical flows are encoded as RGB-like images and tokenized with the same VQ tokenizer as camera frames, so a single autoregressive Transformer can process an interleaved sequence of motion and appearance tokens under a shared vocabulary. Experiments on challenging manipulation benchmarks report state-of-the-art performance with substantially improved sample efficiency, supporting the claim that explicit motion reasoning better bridges pretraining and policy fine-tuning.

**DreamVLA (Zhang et al., 2025b).** DreamVLA recasts VLA as a perception–prediction–action framework by forecasting compact "world knowledge" to support inverse dynamics. Instead of full frame prediction, it predicts targeted cues—dynamic regions (via optical flow), depth maps, and high-level semantic features —to provide concise yet informative guidance for action planning. A block-wise structured attention masks cross-type interactions to prevent leakage and keep representations disentangled, and a diffusion-based transformer decodes actions from shared latents while separating action factors from non-action features. Experiments show strong results in simulation and the real world, including 4.44 average task length on CALVIN ABC→D and 76.7% real-robot success. Ablations indicate dynamic-region forecasting contributes the largest gains, with depth and semantics offering smaller, complementary benefits.

**F1 (Lv et al., 2025).** F1 integrates goal-conditioned visual foresight into the perception–action loop and frames control as foresight-guided inverse dynamics. The model adopts a Mixture-of-Transformer with three experts for understanding, foresight generation, and action execution. A progressive attention scheme regulates information flow, and the generation expert uses a next-scale prediction mechanism to produce explicit planning targets. Training follows a three-stage recipe: Stage I aligns the generation expert with a pretrained understanding expert; Stage II pretrains the full model on large-scale public robot datasets; Stage III post-trains on task-specific data for embodiment adaptation and fine-grained skills. Experiments across simulation and real-world platforms report consistent gains over reactive baselines, with improved robustness and generalization in dynamic and long-horizon tasks.

**RoboVLMs (Liu et al., 2025).** RoboVLMs provides a systematic study and a flexible framework for transferring foundation VLMs into VLAs, aiming to identify effective design choices. The work indicate that continuous-action policy heads perform best and that model choice, architecture, and data strategy all materially affect performance and data efficiency. The study also examines when to leverage cross-embodiment data by contrasting pre-training on Open-X Embodiment, fine-tuning on target domains, and post-training (pre-train then fine-tune). The resulting RoboVLMs achieve strong results in simulation and real-robot settings and generalize to unseen distractors, backgrounds, objects, and novel skill descriptions.

## H  COMPARISON OF DISCRETE AND CONTINUOUS DIFFUSION

In discrete diffusion, the process involves transitioning between discrete states over time. For instance, in text generation, at each diffusion step, a token might be randomly masked or replaced by another token. These transitions are controlled by a predefined schedule or transition matrix, which specifies the probability of moving between states (e.g., retaining a token, replacing it, or masking it). A classic example is the "mask-predict" method used in language models, where tokens gradually emerge during sampling. Discrete models often rely on techniques like absorbing states (where tokens are masked) or uniform transitions between vocabulary items. In contrast, continuous diffusion models gradually add Gaussian noise to the data based on a predefined noise schedule. For example, at each step, the pixel values of an image may be slightly perturbed by noise until it becomes pure noise. During sampling, the model learns to reverse this process by predicting and subtracting the noise at each step. Figure 8 visualizes the differences in the denoising and noising processes between the two approaches.

