# OpenReview forum: "Unified Diffusion VLA: Vision-Language-Action Model via Joint Discrete Denosing Diffusion Process"
_ICLR.cc/2026/Conference — ICLR 2026 Poster_

### Official Review · Reviewer_jgRu · 2025-10-26

**Soundness:** 3
**Presentation:** 3
**Contribution:** 2
**Rating:** 4
**Confidence:** 4

**Summary:**

The paper proposes a unified vision-language-action model that jointly generates future images and predicts actions through a single synchronous denoising trajectory (JD3P), coupling understanding, generation, and acting in a shared tokenized space with hybrid attention. This joint diffusion approach eliminates reliance on external experts and avoids separating generation from control, delivering state-of-the-art performance on CALVIN, LIBERO, and SimplerEnv with 4× faster inference than autoregressive baselines.

**Strengths:**

The paper introduces a novel Joint Discrete Denoising Diffusion Process that enables the generation of image and action tokens.  This formulation effectively bridges the gap between perception and control, achieving intrinsic synergy across modalities.

Methods: By constructing a unified tokenized multimodal space and employing a **hybrid attention mechanism** (bidirectional within modalities, causal across modalities), the model design is principled and well-grounded.

Experiments: The experiments on simulation are comprehensive (across 3 simulation benchmarks).

**Weaknesses:**

1. **Limited theoretical depth:** The motivation for JD3P is intuitively explained, yet the heoretical justification of why joint denoising leads to superior cross-modal alignment remains shallow. A more formal analysis or deeper ablation would strengthen the claim.
2. **Fairness of comparisons:** Some baselines may differ in model capacity or data scale, and these confounding factors are not fully controlled, potentially exaggerating UD-VLA’s relative gains (**more apple-to-apple ablation baselines are needed**).
3. **Overly dense presentation:** The paper is lengthy and symbol-heavy. Some equations (e.g., βt, κt) appear abruptly, requiring readers to backtrack for context.

**Questions:**

1. The single-step mask-prediction loss (Eq. 8) might limit temporal dependency modeling. Have multi-step noise-schedule variants been explored?
2. A key question is whether JD3P’s synchronous denoising is dependent on the prediction tasks and backbone architecture. To disentangle this, I suggest: 1 Porting JD3P to an alternative backbone and repeating the training/evaluation to assess transferability; and
2 Performing controlled comparisons under the same backbone and training protocol against fully autoregressive and partially autoregressive baselines (e.g., VQ-based image generation followed by action decoding).
These experiments would help determine whether improvements stem from the synchronous denoising mechanism per se, or from the image-prediction tasks/backbone, thereby isolating JD3P’s contribution.
3. More details of setups, demos, and experiments in the real world are required.

---

> ### Author Response · Authors · 2025-11-22
> **Response to Reviewer jgRu (1/4)**
>
> We deeply appreciate your insightful comments and efforts in reviewing our manuscript. We respond to each of your comments one by one in what follows. In the revised main manuscript, we mark our major revisions as “blue”.
>
> **[Weakness 1] Limited theoretical depth: The motivation for JD3P is intuitively explained, yet the heoretical justification of why joint denoising leads to superior cross-modal alignment remains shallow. A more formal analysis or deeper ablation would strengthen the claim.**
>
> **Reply:** **Theoretical analysis.** We have provided a theoretical analysis of JD3P in Section 3.1. Specifically, the noising image tokens and action tokens are always in the same denoising timesteps. The action tokens in each timestep are conditioned on the image tokens at the same timestep, which ensures the effective and structured information transmission in the feature space.
>
> **Deeper Ablation.** To further prove our theoretical analysis, we have conducted deeper ablation studies. If we understand correctly, this part is similar to your Question 2. Please refer to our reply to your Question 2.
>
> **[Weakness 2] Fairness of comparisons: Some baselines may differ in model capacity or data scale, and these confounding factors are not fully controlled, potentially exaggerating UD-VLA’s relative gains (more apple-to-apple ablation baselines are needed).**
>
> **Reply:** We agree that fair, apple-to-apple comparisons are important. Following your suggestions, we carefully match the **data scale, model capacity, training batch size, number of optimization steps, and evaluation protocol** with UniVLA. UniVLA is also built on the Emu3  and includes a world-model post-training stage, thus it is suitable for apple-to-apple comparisons.
>
> As shown in the table below, under strictly controlled settings, UD-VLA consistently outperforms UniVLA. Moreover, when we extend UniVLA with outputting future image prediction as visual chain-of-thought (CoT) during inference, as ours, its performance actually degrades. This is because autoregressively predicting visual tokens before action tokens may lead to error accumulation. In contrast, our JD3P design mitigates this issue by jointly predicting future images and actions.
>
> | Method                                            |   1/5 | 2/5 |  3/5 | 4/5 |  5/5 | Average Length |
> | :------------------------------------------------ | :---: | :---: | :---: | :---: | :---: | :-------: |
> | UniVLA                                            | 0.960 | 0.904 | 0.848 | 0.800 | 0.732 |   4.24    |
> | UniVLA + Future Image Prediction During Inference | 0.964 | 0.904 | 0.828 | 0.772 | 0.752 |   4.18    |
> | UD-VLA                                            | 0.992 | 0.968 | 0.936 | 0.904 | 0.840 |   4.64    |
>
> **[Weakness 3] Overly dense presentation: The paper is lengthy and symbol-heavy. Some equations (e.g., βt, κt) appear abruptly, requiring readers to backtrack for context.**
>
> **Reply**: Thank you for this suggestion. We have deleted unnecessary symbols, i.e., $κ_t$, and simplified several parts of the equation to make the symbols less dense，i.e., Eq. (12) . Regarding the need to backtrack, note that $\beta_t$ in Eq. (3) and $\beta$ in Eq. (8) originally denoted different quantities. To avoid this notational clash, we have renamed the weighting coefficient $\beta$ in Eq. (8) to $\omega$.

---

> ### Author Response · Authors · 2025-11-22
> **Response to Reviewer jgRu (2/4)**
>
> **[Question 1] The single-step mask-prediction loss (Eq. 8) might limit temporal dependency modeling. Have multi-step noise-schedule variants been explored?**
>
> **Reply**: We follow prior work [1,2,3,4] and train our discrete diffusion model with a **single-step mask-prediction** objective. Following the reviewer’s suggestion, we additionally conduct an ablation on **multi-step noise-schedule variants**.
>
> In the original setting, each training sample is corrupted at **one** randomly sampled timestep, and we apply a single masked Cross-Entropy loss. In the multi-step setting, for the *same* sample, we choose multiple timesteps from the cosine scheduler, create several corrupted versions with different noise levels within one batch, and compute the loss for each version separately. We keep the effective batch size and the total number of optimization steps fixed across all variants.
>
> As shown in the Table below, the performance of the multi-step noise schedule is almost identical to that of the single-step mask-prediction strategy. This indicates that UD-VLA is largely insensitive to the noise schedule. The reason is that, given enough training iterations, even with a single-step mask-prediction loss, the model will encounter different corruption levels of the same sample over time.
>
> | Scheduling strategy |  1/5 |  2/5 | 3/5 | 4/5 |  5/5   | Average Length |
> | :-----------------: | :---: | :---: | :---: | :---: | :---: | :-------: |
> |       1-step        | 0.992 | 0.968 | 0.936 | 0.904 | 0.840 |   4.64    |
> |       4-step        | 0.976 | 0.958 | 0.928 | 0.892 | 0.854 |   4.61    |
> |       8-step        | 0.992 | 0.960 | 0.920 | 0.896 | 0.856 |   4.62    |
>
> [1] Jiaxin Shi, Kehang Han, Zhe Wang, Arnaud Doucet, and Michalis K. Titsias. Simplified and generalized masked diffusion for discrete data, NIPS 2024.
>
> [2] Ishaan Gulrajani and Tatsunori B Hashimoto. Likelihood-based diffusion language models, NIPS 2023.
>
> [3] Jiacheng Ye, Zhihui Xie, Lin Zheng, Jiahui Gao, Zirui Wu, Xin Jiang, Zhenguo Li, and Lingpeng Kong. Dream 7b: Diffusion large language models, Arxiv 2025.
>
> [4] Zhihui Xie, Jiacheng Ye, Lin Zheng, Jiahui Gao, Jingwei Dong, Zirui Wu, Xueliang Zhao, Shansan Gong, Xin Jiang, Zhenguo Li, et al. Dream-coder 7b: An open diffusion language model for code, Arxiv 2025.

---

> ### Author Response · Authors · 2025-11-22
> **Response to Reviewer jgRu (3/4)**
>
> **[Question 2] A key question is whether JD3P’s synchronous denoising is dependent on the prediction tasks and backbone architecture. To disentangle this, I suggest: 1 Porting JD3P to an alternative backbone and repeating the training/evaluation to assess transferability; and 2 Performing controlled comparisons under the same backbone and training protocol against fully autoregressive and partially autoregressive baselines (e.g., VQ-based image generation followed by action decoding). These experiments would help determine whether improvements stem from the synchronous denoising mechanism per se, or from the image-prediction tasks/backbone, thereby isolating JD3P’s contribution.**
>
> **Reply**: Following your 2 suggestions, we conduct two sets of experiments:
>
> **1**. We port JD3P to an alternative backbone (*i.e.* Qwen2.5-VL [1]) in the table below. To incorporate our JD3P on it, we first reformulate the autoregressive decoding into our discrete diffusion decoding, which improves the overall performance. Furthermore, applying JD3P significantly improved the average length. This demonstrates that our method has transferability and can be applied to different backbones.
>
> |  Backbone  |  Image Generation  | Action Generation  | JD3P | Average Length |
> | :--------: | :----------------: | :----------------: | :--: | :-------: |
> | Qwen2.5-VL |   Autoregressive   |   Autoregressive   |  ❌   |   3.75    |
> | Qwen2.5-VL | Discrete Diffusion | Discrete Diffusion |  ❌   |   3.89    |
> | Qwen2.5-VL | Discrete Diffusion | Discrete Diffusion |  ✅   |   4.20    |
>
> **2**. As shown in the table below, we perform controlled comparisons under the same backbone (*i.e.* Emu3 [2]) and training protocol against fully autoregressive and partially autoregressive baselines. Specifically:
> (1) **Image generation as an auxiliary task**. When we completely remove the image generation objective from training, the model attains the worst performance. Once the image generation task is incorporated—regardless of the specific formulation—the average horizon length (i.e., the length of successful action sequences) improves substantially.
> (2) **Autoregressive *v.s.* discrete diffusion**. Given the presence of the auxiliary image generation task, the results in the table show that the fully discrete diffusion variant outperforms the partially autoregressive variant, which in turn outperforms the fully autoregressive variant. This indicates that, under discrete tokenization, diffusion-based generation enjoys an inherent advantage over purely autoregressive decoding, and further validates that a fully diffusion-based design attains the best performance.
> (3) **JD3P**. After establishing the superiority of the fully diffusion-based scheme, we further compare separate denoising and joint denoising (JD3P). The results demonstrate that JD3P yields a substantial performance gain over separate denoising.
> Taken together, this comprehensive set of experiments shows that our performance improvements stem from all key components: introducing image generation as an auxiliary task, adopting a fully discrete diffusion framework, and employing JD3P for joint denoising.
>
> | Backbone |  Image Generation  | Action Prediction  | JD3P | Average Length |
> | :------: | :----------------: | :----------------: | :--: | :-------: |
> |   Emu3   |         ❌          |   Autoregressive   |  ❌   |   3.78    |
> |   Emu3   |   Autoregressive   |   Autoregressive   |  ❌   |   4.18    |
> |   Emu3   | Discrete Diffusion |   Autoregressive   |  ❌   |   4.24    |
> |   Emu3   |   Autoregressive   | Discrete Diffusion |  ❌   |   4.21    |
> |   Emu3   | Discrete Diffusion | Discrete Diffusion |  ❌   |   4.35    |
> |   Emu3   | Discrete Diffusion | Discrete Diffusion |  ✅   |   4.64    |
>
> [1] Bai et al., Qwen2.5-VL Technical Report, Arxiv 2025
>
> [2] Wang et al., Emu3: Next-Token Prediction is All You Need, Arxiv 2025

---

> ### Author Response · Authors · 2025-11-22
> **Response to Reviewer jgRu (4/4)**
>
> **[Question 3] More details of setups, demos, and experiments in the real world are required.**
>
> **Reply**: At first, we overview our experimental settings in the Appendix Section C and Figure 5. Our experimental tools include target objects and distractors. Target objects are the ones the robot is expected to interact with, while distractors are used to interfere with the model's actions. We further divided the objects and backgrounds into seen ones and unseen ones. The unseen ones are used for the unseen evaluation of "putting blocks".
>
> Second, as shown in Figure 6 and Section D of the appendix of the revised main manuscripts, we provide demos of all tasks and clearly define them:
> - Stacking Bowls: The robot is expected to lift one bowl and stack it on the other.
> - Putting Blocks: The robot is expected to lift the target block and put it into the box.
> - Flipping Towers: The robot is expected to grasp one side of the tower and move to the other side, until the towels are folded.
>
> For experimental details, we train our model for real-world experiments on 4 NVIDIA H100 GPUs for 24 hours, a total of 9000 steps. The batch size is 64, chunk size is 8, and the learning rate is 8e-5 with weight decay of 0.1.

---

> > ### Comment · Reviewer_jgRu · 2025-11-26
> >
> > The authors have addressed my concerns, and I will increase my rating to 6.

---

> ### Author Response · Authors · 2025-11-26
>
> Dear Reviewer jgRu,
>
> We are happy to hear that our response addressed your concerns well! Also, we appreciate your support for our work. If you have any further questions or suggestions, please do not hesitate to let us know.
>
> Best regards,
>
> Authors

---

### Official Review · Reviewer_jYyP · 2025-10-27

**Soundness:** 3
**Presentation:** 2
**Contribution:** 2
**Rating:** 4
**Confidence:** 3

**Summary:**

In this work, the authors propose joint denoising of a subsequent visual frame and the action to take using discrete diffusion over a unified token space, with a causal attention mechanism such that the visual generation does not reference the action generation.  The authors report results over a variety of simulated settings as well as a real world setting.  Speed-up is done by KV-caching and fixing identifier token positions.

**Strengths:**

The strengths of this paper lie in its engineering for fast inference (the authors report 4x faster than autoregressive techniques).  Furthermore, the hybrid attention mechanism is interesting, although it remains to be seen if it is indeed true that such a bias helps decision-making by adding thorough comparisons against visual planning techniques that also generate actions conditioned on future visual info (but future visual info generation is not conditioned on actions), as well as joint generation techniques that do not have this causal restriction (UVA, PAD).  This is elaborated further below.

**Weaknesses:**

The real-world setting was not elaborated upon properly; it is not obvious what objects are considered seen or unseen or what scenes are considered novel.  Appendix Section C does not offer any clarifying details beyond the information provided in Figure 3.  There are no visualizations for any of the tasks listed of "stacking bowls, putting blocks (into a box) and flipping towers"; rather the only visualization in Section B is an unrelated task of "grasp the pink block".  Simply grasping a block is quite different from pick-and-place.  Thorough details, visualizations, and demos of the tasks are necessary - particularly what novelty means.

This reviewer believes there is a large body of prior research that is relevant but missing from the submission.  The work seems to be premised off the idea that generating a future image first then decoding the intermediate action through inverse kinematics helps (Section 4.3, Lines 418-424).  This suggests that visual planning works, which explicitly generate future visual frames and then convert them into actions to execute through an inverse dynamics model, are quite relevant to compare against.  There is a rich body of work that investigates this approach [1,2,3,4] - which aligns with the authors' finding that generating a future image (or videos) before generating the action should help with decision-making.  Referencing them and even comparing against them; naturally, one may expect performance speed to be better for UD-VLA/JD3P with joint decoding; but success rate performance comparisons would be still interesting to report.

Separately, there are works that also seek to denoise subsequent images and actions jointly (UVA [5], PAD [6]).  It would be interesting to also compare against them, as they directly perform joint generation of action and images as in this work.  In this context, perhaps more emphasis can be put on the causal attention mechanism that enforces image generation not to condition on action generation even during the joint denoising process (which once again spiritually relates to visual planning approaches) - to differentiate from these joint vision-action denoising approaches.

To summarize, the joint decoding part reminds readers of other joint image-action decoding techniques [5,6] but the fact that the authors chose to do so with causal attention reminds readers of visual planning techniques[1,2,3,4].  Combined, there is novelty in the proposed approach, but background works and comparisons should still be done for thoroughness.

[1] Wang et al., Learning Robotic Manipulation through Visual Planning and Acting, RSS 2019.

[2] Du et al., Learning Universal Policies via Text-Guided Video Generation, NeurIPS 2023.

[3] Du et al., Video Language Planning, ICLR 2024.

[4] Luo et al., Solving New Tasks by Adapting Internet Video Knowledge, ICLR 2025.

[5] Li et al., Unified Video Action Model, RSS 2025.

[6] Guo et al., Visual Policy Learning via Joint Denoising Process, NeurIPS 2024.

**Questions:**

In visual planning literature, IDM design $p(a \mid s, s')$ is quite sensitive to finding an appropriate frame-skip/horizon $h$ such that $s'$ is $h$ timesteps away from $s$.  Sometimes a more aggressive future frame allows more useful information in predicting action $a$, especially if the rendering frequency is high (and there is lots of redundancy between $s$ and $s'$).  In this work, is $h$ always chosen to be 1 (is the subsequent frame generated always the temporally next frame)?  What choice for $h$ makes the method perform the best and under which environments/tasks?  Can you perform an ablation table over different choices of $h$?

---

> ### Author Response · Authors · 2025-11-22
> **Response to Reviewer jYyP (1/3)**
>
> We deeply appreciate your insightful comments and efforts in reviewing our manuscript. We respond to each of your comments one by one in what follows. In the revised main manuscript, we mark our major revisions as "blue".
>
> **[Weakness 1] The real-world setting was not elaborated upon properly; it is not obvious what objects are considered seen or unseen or what scenes are considered novel. Appendix Section C does not offer any clarifying details beyond the information provided in Figure 3. There are no visualizations for any of the tasks listed of "stacking bowls, putting blocks (into a box) and flipping towers"; rather the only visualization in Section B is an unrelated task of "grasp the pink block". Simply grasping a block is quite different from pick-and-place. Thorough details, visualizations, and demos of the tasks are necessary - particularly what novelty means.**
>
> **Reply**: **1. Unseen Settings**: We have supplemented the details of our real-world setup in the Appendix Section C and Figure 5 of the revised main manuscript. As shown in Figure 5, we provide an overview of our systems, including target objects that the robot is expected to interact with, and distractors. The objects are divided into seen ones that exist in the training sets and unseen ones that are used for the unseen evaluation of "putting blocks". The backgrounds are divided into seen and unseen, where the unseen background is used for the unseen evaluation of "putting blocks".
>
> **2. Task Visualization and Demo**: We have supplemented the visualization and definitions of all real-world tasks in Figure 6 and Appendix Section D of the revised main manuscript:
> - Stacking Bowls: The robot is expected to lift one bowl and stack it on the other.
> - Putting Blocks: The robot is expected to lift the target block and put it into the box.
> - Flipping Towers: The robot is expected to grasp one side of the tower and move to the other side, until the towels are folded.

---

> ### Author Response · Authors · 2025-11-22
> **Response to Reviewer jYyP (2/3)**
>
> **[Weakness 2] This reviewer believes there is a large body of prior research that is relevant but missing from the submission. The work seems to be premised off the idea that generating a future image first then decoding the intermediate action through inverse kinematics helps (Section 4.3, Lines 418-424). This suggests that visual planning works, which explicitly generate future visual frames and then convert them into actions to execute through an inverse dynamics model, are quite relevant to compare against. There is a rich body of work that investigates this approach [1,2,3,4] - which aligns with the authors' finding that generating a future image (or videos) before generating the action should help with decision-making. Referencing them and even comparing against them; naturally, one may expect performance speed to be better for UD-VLA/JD3P with joint decoding; but success rate performance comparisons would be still interesting to report. Separately, there are works that also seek to denoise subsequent images and actions jointly (UVA [5], PAD [6]). It would be interesting to also compare against them, as they directly perform joint generation of action and images as in this work. In this context, perhaps more emphasis can be put on the causal attention mechanism that enforces image generation not to condition on action generation even during the joint denoising process (which once again spiritually relates to visual planning approaches) - to differentiate from these joint vision-action denoising approaches. To summarize, the joint decoding part reminds readers of other joint image-action decoding techniques [5,6] but the fact that the authors chose to do so with causal attention reminds readers of visual planning techniques[1,2,3,4]. Combined, there is novelty in the proposed approach, but background works and comparisons should still be done for thoroughness.**
>
> **Reply:** We sincerely appreciate your comprehensive pointers to additional prior work. They are highly valuable, and we have incorporated discussions and citations of all these works into **the related work section of the revised main manuscript**. In our work, we adopt a unified paradigm that integrates visual generation and inverse kinematics models [1, 2, 3, 4] within a single VLA framework, where the visual generation component serves as a form of chain-of-thought. Such a unified model facilitates more effective information flow across different modalities. We select the most impactful open-source method, UniPi, for comparison on the CALVIN benchmark. The results show that our method consistently outperforms Unipi across all reported metrics. In the table below, SR denotes the Success Rate of each sub-task in a row.
>
> | Method | SR 1/5 | SR 2/5 | SR 3/5 | SR 4/5 | SR 5/5 | Average Length |
> | :----: | :----: | :----: | :----: | :----: | :----: | :------------: |
> | Unipi  | 0.575  | 0.154  | 0.083  | 0.078  | 0.041  |      0.93      |
> |  Ours  | 0.992  | 0.968  | 0.936  | 0.904  | 0.840  |      4.64      |
>
> We agree that our method shares similarities with approaches that jointly denoise subsequent images and actions (UVA [5], PAD [6]), and we appreciate your observation that the use of causal attention is one key distinguishing factor. In addition, to the best of our knowledge, we are the first to scale such joint denoising from relatively small DiT-style models to large language models with more than 7B parameters, achieving further performance gains (as shown in the corresponding tables and analyses). We believe that this line of scaling study will help advance future research on joint image–action generation. We also compare our method with the strong open-source model UVA on the LIBERO-long benchmark, and the results demonstrate that our method is competitive:
>
> | Method | Success Rate |
> | :----: | :----------: |
> |  UVA   |     0.90     |
> |  Ours  |     0.95     |
>
> [1] Wang et al., Learning Robotic Manipulation through Visual Planning and Acting, RSS 2019.
>
> [2] Du et al., Learning Universal Policies via Text-Guided Video Generation, NeurIPS 2023.
>
> [3] Du et al., Video Language Planning, ICLR 2024.
>
> [4] Luo et al., Solving New Tasks by Adapting Internet Video Knowledge, ICLR 2025.
>
> [5] Li et al., Unified Video Action Model, RSS 2025.
>
> [6] Guo et al., Visual Policy Learning via Joint Denoising Process, NeurIPS 2024.

---

> ### Author Response · Authors · 2025-11-22
> **Response to Reviewer jYyP (3/3)**
>
> **[Question] In visual planning literature, IDM design $p(a \mid s, s')$ is quite sensitive to finding an appropriate frame-skip/horizon $h$ such that $s'$ is $h$ timesteps away from $s$. Sometimes a more aggressive future frame allows more useful information in predicting action $a$, especially if the rendering frequency is high (and there is lots of redundancy between $s$ and $s'$). In this work, is $h$ always chosen to be 1 (is the subsequent frame generated always the temporally next frame)? What choice for $h$ makes the method perform the best and under which environments/tasks? Can you perform an ablation table over different choices of $h$?**
>
> **Reply**：First, we apologise if our previous description was misleading and may have caused confusion. We do **not** simply fix $h = 1$. Instead, we set $h$ equal to the action chunk sizes. When predicting an $h$-step action sequence, we use the $h$-th future frame as the IDM condition $s'$. The $h$ matches the number of frames contained in one second for each dataset to provide a stable training across different datasets. In our context, we set $h = 10$ for CALVIN, $h = 10$ for LIBERO, $h = 5$ for SimpleEnv, and $h = 8$ for the real-robot setup.
>
> To further address the reviewer’s concern, we conduct an ablation over different choices of decoding horizon on CALVIN ABCD$\rightarrow$D. As shown in the Table below, a decoding horizon of 10 achieves the best performance, which is consistent with our design choice and analysis. In the table below, SR denotes the Success Rate of each sub-task in a row.
>
> | Decoding horizon (h) |  SR 1/5 | SR 2/5 | SR 3/5 | SR 4/5 | SR 5/5  | Average Length |
> | :------------------: | :---: | :---: | :---: | :---: | :---: | :------: |
> |          5           | 0.976 | 0.904 | 0.864 | 0.824 | 0.752 |  4.320   |
> |          8           | 0.984 | 0.928 | 0.896 | 0.840 | 0.768 |  4.416   |
> |      10 (ours)       | 0.992 | 0.968 | 0.936 | 0.904 | 0.840 |  4.640   |
> |          20          | 0.968 | 0.904 | 0.824 | 0.800 | 0.760 |  4.356   |

---

### Official Review · Reviewer_8ozA · 2025-11-01

**Soundness:** 4
**Presentation:** 4
**Contribution:** 3
**Rating:** 8
**Confidence:** 2

**Summary:**

This paper presents Unified Diffusion VvLA, a novel VLA method that tightly integrates understanding, generation, and action through a Joint Discrete Denoising Diffusion Process (JD3P). Unlike prior approaches that separate visual generation and action prediction, UD-VLA performs joint multimodal denoising in a unified tokenized space, enabling synchronous optimization of image and action generation. The authors propose a hybrid attention mechanism, a two-stage training pipeline, and inference-time optimizations to balance accuracy and efficiency. Extensive experiments on CALVIN, LIBERO, and SimplerEnv benchmarks, as well as real-world robot evaluations, demonstrate state-of-the-art (SOTA) performance and 4× faster inference compared to autoregressive baselines

**Strengths:**

Overall, I think that paper is a timely contribution that significantly advances the state of the art. The paper demonstrates clear innovation, rigorous experimentation, and practical relevance.

- JD3P unifies multimodal generation and control within a single denoising trajectory, representing a conceptual and technical advance over modular or autoregressive VLAs.
- UD-VLA achieves SOTA on multiple simulation and real-world benchmarks, consistently outperforming strong baselines such as UniVLA, F1, and DreamVLA, with notable efficiency gains (4× faster decoding).
- The proposed attention structure effectively balances intra-modal richness with inter-modal causality, enabling actions to be grounded in predicted future observations.
- The paper includes well-designed ablation studies that clarify the contributions of hybrid attention, future image generation, and joint denoising.
- Demonstrating strong generalization to unseen environments and objects enhances the paper’s practical impact and credibility.
- Despite the technical depth, the paper is well-structured, making complex ideas accessible to both robotics and multimodal learning audiences.

**Weaknesses:**

I do not think that paper has major weaknesses. Some minor ones:

- While the appendix mentions that generated images lack fine-grained fidelity, the main text could better discuss potential failure cases or constraints (e.g., scalability to high-resolution images or multi-robot domains).

**Questions:**

I don’t have additional questions.

---

> ### Author Response · Authors · 2025-11-22
> **Response to Reviewer 8ozA**
>
> Thank you for your careful reading and valuable feedback. We are encouraged by your recognition. We respond to your comments and talk about the constraints in detail in the following paragraph.
>
> **[Weakness] While the appendix mentions that generated images lack fine-grained fidelity, the main text could better discuss potential failure cases or constraints (e.g., scalability to high-resolution images or multi-robot domains).**
>
> **Reply**：
> Although our generated images are already close to the real observations and are enough for basic robotic manipulation tasks, we would like to discuss the cases considering more fine-grained images and tasks.
>
> Specifically, we additionally design a more challenging and fine-grained task that requires the robot to grasp the tiny toy (please see Figure 7 of the revised main manuscript). We tackle this challenge with more fine-grained visual tokenizations. Specifically, we compare our original setting (625 visual tokens) and a more fine-grained version (1024 visual tokens). As shown in the table below, increasing the number of visual tokens yields a slight improvement in task success rate. This indicates that for more fine-grained tasks or higher-resolution images, our methods have the potential to be improved through more fine-grained visual tokenizations, while our present choice is a balance of accuracy and efficiency.
>
>
> | Visual tokens | Success rate (%) | Inference time (s) |
> | :-----------: | :--------------: | :----------------: |
> |     625     |       79.0       |        1.75        |
> |     1024     |       83.0       |        2.74        |

---

### Official Review · Reviewer_qxoz · 2025-11-01

**Soundness:** 4
**Presentation:** 4
**Contribution:** 4
**Rating:** 6
**Confidence:** 3

**Summary:**

The paper presents a novel approach to build Vision-Language-Action(VLA) models. It tackles the current challenges that current methods often treat visual prediction and action prediction as two separate processes, which limits their mutual connection when training.

The authors propose the Unified Diffusion VLA (UD-VLA), a unified vision-language-action model that is built upon a Joint Discrete Denoising Diffusion Process (JD3P). In specific, it maps all modalities (texts, images, actions) into a shared discrete token space and then process with hybrid attention mask.

Then, the authors design a two-stage training pipeline to activate the image generation capabilities by firstly post train the backbone for future-image prediction and then jointly train image and action. The paper reaches SOTA across CALVIN, LIBERO and SimplerEnv. More importantly, they claim four-time faster inference speed than normal autoregressive policies.

**Strengths:**

•	Unified Tokenization and Hybrid Attention: In this paper, all modalities are converted into a single sequence into a single sequence of discrete tokens instead of simply concatenation. Then the paper allows bidirectional attention within single modality but enforces causal attention across modalities. It’s a solid architecture design that naturally frames the task as utilizing futures images with actions but prevent leakage, improving the interpretability. Corresponding ablation study in Table 5,6 shows as a solid prof that hybrid attention is better than causal and simple bidirectional attention. In addition, future-image prediction provides more help than reconstructing the current-frame.

•	Novelty for JD3P: The idea of selectively reconstructs a subset of masked positions while leaving clean others hidden at each denoising step is interesting and therefore JD3P make sure future images and action tokens are generated within the same denoising step. It tackles the problem of weak connection in previous unified VLAs between visual generation and action prediction.

•	Strong Performance on various dataset: The paper shows SOTA performance on CALVIN, LIVERO, and SimplerEnv which proof its strong capabilities in long-horizon reasoning and execution, its strong generalization ability and its real-world transferability.

•	Inference Efficiency: In Table 7, the paper demonstrates its 4 times faster compared to autoregressive methods and matin its performance advantage, which is significant.

**Weaknesses:**

•	Visual Fidelity: Like the author also mentioned, the generated future images lack high visual fidelity, likely due to the VQ tokenization and lack of large-scale generative pretraining. While the author claims that it’s sufficient, it’s unclear how low fidelity will impact on specific tasks which required fine-grained visual details in real world settings.

•	Sensitivity for hyperparameter: In Equation 8, the paper introduces β down-weights the visual tokens to avoid their dominance. It remains unclear for the difficulty to tune this without any training analysis.

•	Pretrained VLM: The paper introduces its two-stage design pipeline when training. However, it utilized a recent powerful pretrained VLM which is effective but make it difficult to distinguish between the contribution of performance from JD3P from previous methods.

**Questions:**

•	Figure 1 seems to be a little bit hard to follow especially for its training process and how JD3P works simply by looking at the figure.

•	In Table 7, did the speed up represent the whole end-to-end process or simply the decoding process?

•	How critical is the pretrained VLM working for the backbone? Can you have an additional ablation study, or some explanation clarify the performance is achieved compared to simply using baseline?

---

> ### Author Response · Authors · 2025-11-22
> **Response to Reviewer qxoz (1/2)**
>
> We deeply appreciate your insightful comments and efforts in reviewing our manuscript. We respond to each of your comments one by one in what follows. In the revised main manuscript, we mark our major revisions as "blue".
>
> **[Weakness 1] Visual Fidelity: Like the author also mentioned, the generated future images lack high visual fidelity, likely due to the VQ tokenization and lack of large-scale generative pretraining. While the author claims that it’s sufficient, it’s unclear how low fidelity will impact specific tasks that require fine-grained visual details in real-world settings.**
>
> **Reply**：First, the manipulation tasks that we consider do not require extremely fine-grained visual details. Therefore, we use $25 \times 25 = 625$ visual tokens to represent one image with enough fidelity in our main experiments to balance accuracy and efficiency.
>
> Second, to further address the reviewer’s concern, we additionally design a more challenging task that the robot is required grasp the tiny toy (please see Figure 7 of the revised main manuscript). This task proposes a higher demand for fine-grained visual perception. We tackle this challenge with more fine-grained visual tokenizations, i.e., a larger number of visual tokens. Specifically, we compare $25 \times 25$ (625 tokens) and $32 \times 32$ (1024 tokens). As shown in the table below, increasing the number of visual tokens leads to higher success rates, while the improvement is not obvious. At the same time, more visual tokens substantially increase inference latency because the model needs to infer a longer token sequence. The results prove that our methods can adapt to tasks that require fine-grained through more fine-grained visual tokenization, and our present choice is a balance of accuracy and efficiency.
>
> | Visual tokens | Success rate (%) | Inference time (s) |
> | :-----------: | :--------------: | :----------------: |
> |     25×25     |       79.0       |        1.75        |
> |     32×32     |       83.0       |        2.74        |
>
> **[Weakness 2] Sensitivity for hyperparameter: In Equation 8, the paper introduces β downweights the visual tokens to avoid their dominance. It remains unclear for the difficulty to tune this without any training analysis.**
>
> **Reply**：In our paper, we follow the fine-tuning setup of Emu3 [1] and set the loss weight of the visual tokens to 0.5. To address the reviewer’s concern, we additionally conduct an ablation study on the visual-token loss weight on CALVIN ABCD→D. As shown in the table below,  the average length is not sensitive to the choice of β. Choosing β = 0.5 yields the best overall performance. In the table below, SR denotes the Success Rate of each sub-task in a row.
>
> | β  |  SR 1/5 | SR 2/5 | SR 3/5 | SR 4/5 | SR 5/5  | Average Length |
> | :------------------: | :---: | :---: | :---: | :---: | :---: | :------: |
> |          0.2           | 0.984 | 0.936 | 0.912 | 0.896 | 0.812 |  4.54  |
> |      0.5       | 0.992 | 0.968 | 0.936 | 0.904 | 0.840 |  4.64   |
> |      1.0        | 0.976 | 0.952 | 0.920 | 0.872 | 0.800 |  4.55   |
>
>
> [1] Xinlong Wang, Xiaosong Zhang, Zhengxiong Luo, Quan Sun, Yufeng Cui, Jinsheng Wang, Fan Zhang, Yueze Wang, Zhen Li, Qiying Yu, et al. Emu3: Next-token prediction is all you need. arXiv preprint arXiv:2409.18869, 2024.
>
>
> **[Weakness 3] Pretrained VLM: The paper introduces its two-stage design pipeline when training. However, it utilized a recent powerful pretrained VLM, which is effective but makes it difficult to distinguish between the contribution of performance from JD3P from previous methods.**
>
> **Reply**：We conducted additional ablations on three design choices based on the same pretrained VLM backbone: (i) whether to perform our two-stage training, including the world-model post-training and fine-tuning, and (ii) whether to perform our Joint Discrete Denoising Diffusion Process (JD3P) or simply autoregressive decoding.
>
> The results are shown in the table below. Directly fine-tuning the pretrained VLM (i.e., Emu3) for action prediction yields a low average length of 2.46. This indicates that simply fine-tuning the strong pre-trained VLM into VLA is not enough. After two-stage training, the average length rose to 4.18. This indicates that the post-training on the auxiliary task of future image prediction is effective. Finally, replacing the autoregressive decoding with JD3P, so that future frames and actions are predicted jointly within a single denoising process, brings an obvious gain of 0.36, reaching an average length of 4.64.
>
> Overall, these results show that our proposed designs are the key reasons for the strong performance, rather than simply using the pretrained VLM.
>
> | Two-stage Training | Decoding Method | Average Length |
> | :---------------: | :-------------: | :------------: |
> |         ❌         | Autoregressive  |      2.46      |
> |         ✅         | Autoregressive  |      4.18      |
> |         ✅         |      JD3P       |      4.64      |

---

> ### Author Response · Authors · 2025-11-22
> **Response to Reviewer qxoz (2/2)**
>
> **[Question 1] Figure 1 seems to be a little bit hard to follow, especially for its training process and how JD3P works simply by looking at the figure.**
>
> **Reply**: Thank you for your valuable suggestions. We have reorganized Figure 1 and also rewritten the caption to make it clear, as shown in the **revised main manuscripts**. Specifically, for the training process, we first conduct a post-training in a world-model manner to predict future images. Then, with the world modeling capabilities, we further fine-tune our model to generate both future images and actions. For JD3P, constrained by the Figure 1's limited expressive power, comprehensive details are provided in Section 3.1 of the main manuscript. Briefly, the noising image tokens and action tokens are always in the same denoising timesteps. The action tokens in each timestep are conditioned on the image tokens at the same timestep, which ensures the effective, structured information transmission.
>
> **[Question 2] In Table 7, did the speed up represent the whole end-to-end process or simply the decoding process?**
>
> **Reply**: The speed up in the table represents the decoding process. In our system, the decoding process accounts for more than 95% of the whole end-to-end runtime, and our focus is on improving decoding efficiency. Therefore, we report the speed in the decoding process.
>
> **[Question 3] How critical is the pretrained VLM working for the backbone? Can you have an additional ablation study, or some explanation clarify the performance is achieved compared to simply using baseline?**
>
> **Reply**: If we understand correctly, this question is relevant to Weakness 3 above. Please refer to our replies to Weakness 3.

---

### Author Response · Authors · 2025-12-01
**Summary of Changes for Rebuttal**

Dear Area Chair and Reviewers,

We sincerely thank all reviewers for their thoughtful feedback. Given the recent AC reassignment, we would like to briefly summarize the status of our paper before the score reversion to support the AC’s final decision.

We have provided comprehensive responses to all questions raised by four reviewers.

- Reviewer 8ozA assigned a score of **8** and did not identify any major weaknesses.
- Reviewer jgRu raised the score from 4 to **6** on 26 Nov 2025 at 20:26 (AoE), before the bug of OpenReview became publicly known, which is also consistent with the reviewer's comments.
- Reviewer qxoz assigned a score of **6,** and Reviewer jYyP assigned a score of **4**, but neither has responded. We believe we have substantially addressed their concerns.

Based on the reviewers' comments, we have made several improvements to our paper:

**New Experiments**

- We conducted ablation studies of our 1) JD3P, 2) post-training, 3) image prediction, and 4) hyperparameters (including scheduling strategy, number of visual tokens, loss weight of the visual tokens, and prediction horizons), which demonstrates that the model’s superior performance stems from our proposed design rather than merely from the strength of the backbone. (mentioned by Reviewers qxoz, jYyP, and jgRu)
- We additionally include experiments based on the other backbone, demonstrating the transferability of our method. (mentioned by Reviewer jgRu)
- We further conduct an apple-to-apple comparison with the baseline and show the superiority of our approach. (mentioned by Reviewer jgRu)
- We augment the related work on visual and action co-training, and compare their representative approaches with our method, highlighting the advantages of our model. (mentioned by Reviewer jYyP)

**Clarity Enhancements**

- We clearly define the setup of the real-robot experiments and provide visualizations. (mentioned by Reviewer jYyP and  jgRu)
- We clarify the rationale behind our choice regarding visual fidelity in robotic tasks, striking a balance between performance and efficiency. (mentioned by Reviewers qxoz and 8ozA)
- We re-organize Figure 1 and its caption to improve readability. (mentioned by Reviewer qxoz)
- We refine the notations and formulas. (mentioned by Reviewer jgRu)

Sincerely,

The Authors

---

### Meta-Review · Area_Chair_18tx · 2026-01-07

**Summary:**

This paper proposes Unified Diffusion VLA (UD-VLA), a unified vision–language–action framework that jointly denoises future image tokens and action tokens through JD3P in a shared discrete token space, combined with a hybrid attention design (bidirectional within each modality and causal across modalities). The core goal is to more tightly couple visual prediction with action prediction while significantly improving inference efficiency.

Overall, reviewers found the work timely and technically solid. The paper demonstrates strong empirical performance, achieving state-of-the-art results on CALVIN, LIBERO, and SimplerEnv. Reviewer 8ozA was strongly positive (score 8). Reviewer qxoz was positive and near the acceptance threshold (score 6), with most concerns focused on clarification and ablation rather than fundamental issues. Reviewer jgRu initially raised concerns about fairness of comparisons, theoretical depth, and presentation clarity, but after the rebuttal indicated that these concerns were addressed and increased the score to 6. Reviewer jYyP remained the main detractor (score 4), primarily emphasizing insufficient clarity in the real-world setup and missing comparisons and related work; however, the rebuttal substantially expanded task definitions, visualizations, and comparisons (e.g., UniPi, UVA), partially addressing these points.

**Reviewer Concerns:**

The rebuttal meaningfully strengthened the submission. The authors added extensive new experiments and ablations; On the clarity side, the authors expanded the description of the real-robot setup, added task visualizations, clarified the visual fidelity vs. efficiency tradeoff, reorganized Figure 1, and refined notations throughout the paper.

Taken together, the rebuttal directly addresses the most important concern raised during review. The remaining weaknesses are largely about scope and positioning, rather than fundamental technical flaws. Visual fidelity remains limited, but the authors provide concrete evidence that increasing token granularity improves task success at the cost of latency, and they clearly justify their chosen operating point on the accuracy–efficiency tradeoff curve. Given the updated record, there is now clear majority reviewer support (scores of 8, 6, 6 vs. 4). The additional experiments significantly reduce the main confounds raised in the initial reviews, and the remaining concerns do not outweigh the demonstrated empirical strength and practical impact of the work.

**Reviewer Scores:**

Reviewer jgRu: Increased from 4 to 6, as explicitly stated by the reviewer after reading the rebuttal and additional experiments. Others may remain unchanged.

---

### Decision · Program_Chairs · 2026-01-26

Accept (Poster)